


# Statistical and neural network assessment of climatological features of fog and mist at Pula airport in Croatia: from local to synoptic scale

Marko Zoldoš*[1,2], Tomislav Džoić*[3], Jadran Jurković[2], Frano Matić[4], Sandra Jambrošić[2], Ivan Ljuština[2], Maja Telišman Prtenjak[5]

[1]Risk Management Division, Erste & Steiermärkische Bank d.d., Rijeka 51000, Croatia (ORCID: 0009-0005-1117-5716)
[2]Aviation Meteorology department, Croatia Control Ltd., Velika Gorica 10410, Croatia
[3]Laboratory of Physical Oceanography, Institute of Oceanography and Fisheries, Split 21000, Croatia
[4]Department of Marine Studies, University of Split, Split 21000, Croatia (ORCID: 0000-0003-0392-4172)
[5]Department of Geophysics, Faculty of Science, University of Zagreb, Zagreb 10000, Croatia (ORCID: 0000-0002-4941-8278)

*Leading author: Marko Zoldoš, correspondence to: Tomislav Džoić (dzoic@izor.hr)

**Abstract.** A study was conducted on the climatological characteristics of fog and mist at Pula Airport in the northeastern Adriatic, using statistical and machine learning approaches. The study utilized meteorological data from Pula Airport, along with satellite sea surface temperature (SST) data from two coastal areas west and east of the airport, to gain insights into the influence of sea temperature on fog formation. To identify weather patterns associated with the occurrence of fog and mist, wind and mean sea-level pressure (MSLP) data from the ERA5 reanalysis were analyzed using Growing Neural Gas (GNG), a machine learning method. A notable finding was a declining trend in the frequency of fog and mist at the airport, which can be linked to the results of the GNG analysis of the ERA5 data. This analysis showed a decrease in synoptic patterns favorable for fog and mist. Fog occurs mainly between October and March and is primarily associated with weak easterly and northeasterly winds. Additionally, fog is more likely to occur when the sea surface temperature is higher than the air temperature. Mist has similar characteristics to fog, although it is more likely to occur with easterly winds.



## 1 Introduction

According to the World Meteorological Organization (WMO), fog is defined as the suspension of extremely tiny water droplets or ice crystals in the atmosphere, often at a microscopic scale. This natural occurrence significantly reduces visibility on the Earth's surface to less than 1 km (WMO, 1966). The formation of these minuscule water droplets and ice crystals is influenced by various factors, including cooling, increased humidity, and the mixing of air masses with different temperatures (Gultepe, 2007). Mist is a closely related phenomenon, occurring when horizontal visibility at the surface is between 1 and 10 km. In aviation meteorology, mist specifically refers to conditions where surface horizontal visibility is between 1 and 5 km. Fog is a unique atmospheric phenomenon confined to the atmospheric boundary layer (ABL), the lowest part of the atmosphere, whose behavior and characteristics are directly influenced by contact with the Earth's surface. The formation and dissipation of fog are profoundly impacted by synoptic and mesoscale conditions, as well as surface features such as moisture sources (oceans, lakes, rivers), vegetation type, orography, urban areas, and sea currents.

This study aims to investigate the occurrence of fog over an extended period at Pula Airport (44.89°N, 13.92°E), located in the coastal region of Croatia in the northeastern Adriatic (Figure 1). The Adriatic Sea is a large semi-enclosed sea separating the Apennine Peninsula from the Balkans. It is the northernmost arm of the Mediterranean Sea, extending from the Strait of Otranto (where it connects to the Ionian Sea and the rest of the Mediterranean) to the northwest, toward the Po Valley and the Istria Peninsula. This region is prone to marine fog formation due to synoptic-scale effects that can trigger subsidence within the boundary layer, causing stratus clouds to descend to the surface. Similar marine fog events have been observed and analyzed in other regions like the northwestern Pacific and Atlantic Oceans (e.g., Koračin et al., 2001; Koračin and Dorman, 2017). In the Adriatic region, fog is commonly observed between September and May, often disrupting sea transport and port operations (Popović et al., 2014). In addition to affecting sea traffic, fog is a major disruptor of air traffic because of safety, as fog at airports can cause significant flight delays due to poor visibility and low cloud ceilings. These delays result in substantial financial losses for airlines, underscoring the need for accurate fog forecasting. For example, dense fog at New Delhi Airport in India caused losses of approximately 3.9 million U.S. dollars between 2011 and 2016 (Kulkarni et al., 2019). The importance of mitigating these losses is highlighted by a study by Allan (2001), which estimated that improved forecasts during low ceiling/poor visibility events at 3 airports in the New York City area could save up to $240.000 per event.

These factors collectively emphasize the importance of fog research in improving the forecasting process. Unfortunately, the study of fog remains an area of atmospheric science where our understanding is limited, both over land (Gultepe, 2007) and ocean (Koračin and Dorman, 2017). Fog formation requires a fine interplay of processes ranging from synoptic to microscale levels. The typical size of fog condensation nuclei is around 0.1 μm ($10^{-5}$ cm), while the synoptic-scale processes that contribute to fog development occur on a scale of $10^8$ cm or more, making the ratio of interacting length scales about $10^{13}$. The full explanation of fog formation involves various elements, including synoptic conditions (Belo-Pereira and Santos, 2016), land surface characteristics and radiation exchange (Duynkerke, 1991), microphysics (Gultepe and Milbrandt, 2007; Wang et al., 2019), climatology (Stolaki et al., 2009; Veljović et



al., 2015; Belo-Pereira and Santos, 2016), relationships to turbulence intensity (Ju et al., 2020), aerosol (Oztaner and
Yilmaz, 2013), and more. Weather forecasting, the most publicly visible area of meteorology, involves not just
atmospheric science and its applications, but also communication and interaction with users. The forecasting process
always begins with the output from a numerical weather prediction (NWP) model, which, despite continual
development and upgrades, is not perfect and can only offer guidance (Tudor, 2010; Klaić, 2015; Telišman Prtenjak
et al., 2018). An expert forecaster must therefore use both general knowledge of large-scale weather systems and
detailed knowledge of local processes and climatology to refine this guidance into a final forecast product. The
complicated interplay of meteorological parameters that leads to fog or no-fog conditions makes fog forecasting a
significant challenge (Bergot and Koračin, 2021).

This investigation focuses on analyzing the climatological characteristics of fog at Pula Airport (Croatia) and
understanding the general patterns of fog initiation and dissipation. The first goal is to provide detailed statistical
analyses to help understand the local and dynamic processes that lead to fog development. Another important goal is
to gain insight into the influence of sea surface temperature (SST) in the vicinity of the study area on the frequency
and intensity of advective fog, as SST has been shown to have a significant impact on the accuracy of NWP models
(Huang et al., 2022). For this study, an unsupervised machine learning algorithm called the Growing Neural Gas
Network (GNG), a method within artificial neural networks (Martinetz and Schulten, 1991), has been used. The main
aim of applying this method is to classify the synoptic conditions that prevail before and during the occurrence of fog
at Pula into different weather patterns, and to determine if there are patterns that favor fog development. The results
of this research could significantly help local forecasters improve fog forecasting by considering the specific terrain
and coastline features, as well as synoptic and local influences (particularly SST), thereby filling a gap in scientific
knowledge about fog characteristics in this part of the Mediterranean.
**2 Location, data and methods**
**2.1 Location**
Pula Airport is the international airport serving the coastal city of Pula (population 52,411, according to the 2021
census) in western Croatia. It is located approximately 6 km ENE of the city center (Figure 1b). The exact geographical
coordinates of the airport are 44°53'37" N and 13°55'20" E, with an elevation of 84 meters above mean sea level
(AMSL). In 2023, Pula Airport served 413,439 passengers, making it the fifth busiest airport in Croatia by passenger
traffic (source: https://podaci.dzs.hr/2023/en/58556). The airport is situated at the southern tip of the Istrian
Peninsula—the largest peninsula in the Adriatic Sea—positioned between the Gulf of Trieste to the northwest and the
Kvarner Gulf to the east. The climate of Istria is influenced by both the Alpine and Dinaric Alps mountain ranges, as
well as the Mediterranean Sea. Winters in Istria are typically mild and wet, while summers are hot and humid. The
interior of Istria experiences a more continental climate, while the coastal area is significantly influenced by the
Adriatic Sea. Recent research has analyzed bioclimatic parameters, highlighting these climatic boundaries across the
peninsula (e.g., Omazić et al., 2020).



Two weather patterns are most commonly associated with fog in Pula. The first involves a predominantly westerly
flow that advects moist air from the west (W) and northwest (NW) under anticyclonic conditions. In these situations,
advection can occur over a wide geographical area. Fog is often advected from the Po Valley in northern Italy, where
it is a frequent phenomenon during the fall-winter season (Mariani, 2009), across the northern Adriatic to the shores
of Istria. In such weather patterns, fog can sometimes persist for days over the entire affected region (Bendix, 1994).
This is reflected in the fact that Linate Airport in Milan, Italy, was once the European airport with the most annual
shutdowns due to fog (Mariani, 2009). Although advective fog is not as common in the northwestern Adriatic, it still
occurs frequently on the western coast of Istria (Tešić and Brozinčević, 1974), which is climatologically the foggiest
area of the eastern Adriatic (Stipaničić, 1972). The second weather pattern associated with fog in Pula involves a
predominantly easterly flow with advection from the southeast (SE) during a weakening anticyclone. These patterns
are closely related to broader atmospheric circulation over the Adriatic Sea, which is dominated by four main wind
field patterns. It is important to emphasize the influence of the wind regime, which includes the northeasterly (NE)
wind called bora and the southeasterly (SE) wind called sirocco, both typically occurring in the colder part of the year
and influenced by regional synoptic-scale systems. During the warmer part of the year, sea/land breezes and, to a
lesser extent, the Etesian wind are frequent. The change in wind at Pula Airport is influenced by these fundamental
wind regimes (e.g., Pandžić and Likso, 2005; Prtenjak and Grisogono, 2007; Prtenjak et al., 2010; Belušić et al., 2018).

The terrain surrounding the airport is mostly flat, covered with grass vegetation and small forests. In the immediate
vicinity, there are no notable hills or mountains that could significantly influence the weather and climate of the area.
The central part of the airport, along with the nearby southern surroundings, lies in a very shallow basin, which is
prone to nighttime inversions during calm wind conditions and clear skies (as reported by local forecasters). The sea
is very close to the airport—Pula Bay is just 6 km to the west-southwest, and the open waters of the northern Adriatic
Sea are 10 km away. To the east, the open sea is 7 km away, with the small Bay of Kavran situated just 5 km from
Pula Airport. These factors make the weather at the airport very susceptible to marine influence, even though the
airport itself is not directly on the coast.




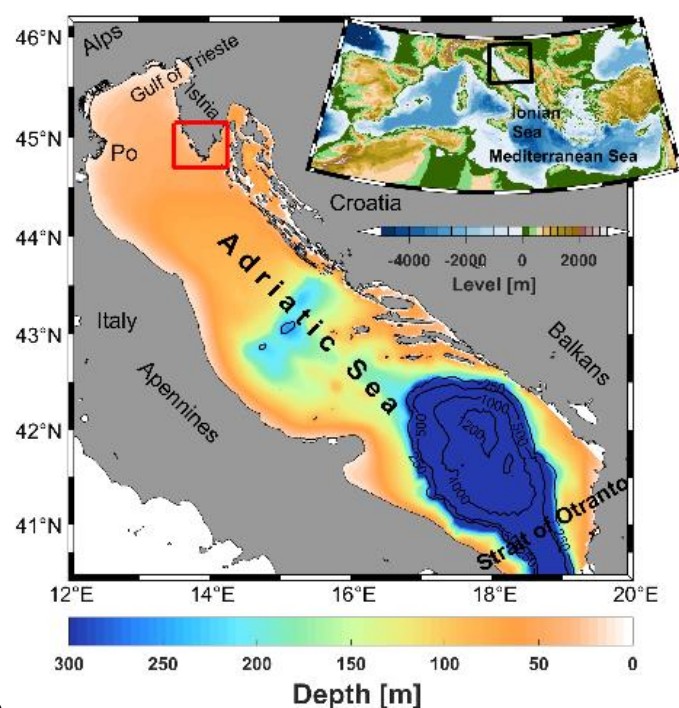

137       a)

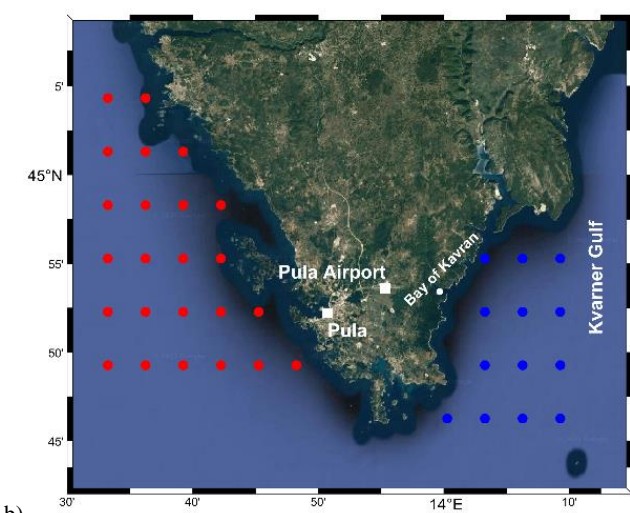

138       b)

**Figure 1. a) Map of the bathymetry of the Adriatic Sea, with the black square marking the area of the Adriatic Sea. The red rectangle on the map marks the immediate surroundings of Pula Airport (b) (© Google Maps 20). The important localities are marked with white squares, while blue and red dots mark the eastern and western grid points from which the satellite SST values were extracted. The wider area of the Mediterranean Sea corresponds to the area of the ERA5 reanalysis. 24**




## 2.2 Data and methods


The dataset used for this study includes half-hourly METAR reports and three-hourly SYNOP reports from the
meteorological station at Pula Airport. The meteorological variables considered are wind speed and direction,
temperature at 2 meters above ground, dew point temperature, relative humidity, surface pressure, cloud cover, and
horizontal visibility. These variables are reported by both station observers and automatic instruments. The dataset
spans a 20-year period from January 1, 2001, to December 31, 2020. All measurements were taken by instruments
located at the airport's meteorological station. Runway 27, the primary runway in use, is equipped with a landing
system capable of Category I operations. This allows for takeoff and landing under low visibility conditions with a
Runway Visual Range (RVR) of up to 550 meters or a ceiling height of 200 feet (approximately 60 meters). This
capability highlights the significant impact of fog on airport operations in Pula—when visibility drops below 550
meters, aircraft landings and takeoffs are not possible.

In addition to data from Pula Airport, daily sea surface temperature (SST) values from the Pula Bay oceanographic
station were utilized. This dataset includes SST measurements taken at 07:00, 14:00, and 21:00 local time from
January 1, 2001, to December 31, 2020. For the same period, reprocessed satellite data of SST in the Mediterranean
region were obtained from the Copernicus Marine Data Store (https://data.marine.copernicus.eu/). This dataset, which
has been optimally interpolated (level 4) with a grid resolution of 0.05° (Merchant et al., 2019), provides a
comprehensive view of sea surface temperatures. Two coastal areas were selected for analysis: one to the west of the
airport towards the open Adriatic Sea and the other to the east towards the Kvarner Gulf (Figure 1). The purpose of
this selection was to extract the average spatial SST for these two regions on a given day and assess their influence on
fog formation. These areas were chosen because they encompass the nearshore sea where the most frequent winds
occur.

To identify the synoptic wind and pressure fields that favored fog occurrence at Pula Airport and across the
Mediterranean region, 10-meter wind and mean sea-level pressure (MSLP) data were sourced from the fifth generation
of ECMWF's ERA5 reanalysis. This reanalysis provides a comprehensive view of global climate and weather from
the last 4 to 7 decades. ERA5 integrates observational data with atmospheric models to deliver detailed and accurate
assessments of past weather. It has a horizontal resolution of 0.25° for latitude and longitude, a temporal resolution of
one hour, and a vertical resolution based on 37 pressure levels (Hersbach et al., 2020a; 2020b). The covered area
extends from 20°W to 40°E and 25°N to 55°N (Figure 1a, smaller map), with data spanning from 1979 to 2019. To
manage the large datasets of 10-meter wind and MSLP and classify them into spatio-temporal patterns that shed light
on atmospheric conditions favoring fog formation at Pula Airport, the Growing Neural Gas Network (GNG) method
was employed. GNG is an unsupervised artificial neural network that clusters high-dimensional input data by reducing
its dimensions and grouping it into best matching units (BMUs) (Fritzke, 1995). Unlike traditional neural networks
with fixed structures, GNG dynamically expands by adding new neurons in response to input patterns. This ability to
grow and adapt allows GNG to effectively cluster data and detect patterns and anomalies. The GNG algorithm has
been successfully used to detect anomalies in Adriatic Sea data, combining various biological and oceanographic



inputs (Šantić et al., 2021; Džoić et al., 2022). In this study, the method outlined in Matić et al. (2022) was applied to
a high-dimensional, 40-year dataset with wind components u and v for the entire year. Nine BMUs per month were
calculated for each month of the year using the GNG algorithm from the Python library NeuPy. Data processing and
visualization were performed using Python and MATLAB, with MATLAB employing the mapping package M_map
(Pawlowicz, 2020) which is available online at www.eoas.ubc.ca/~rich/map.html.
**3 Results and discussion**
**3.1 Climatological analysis**

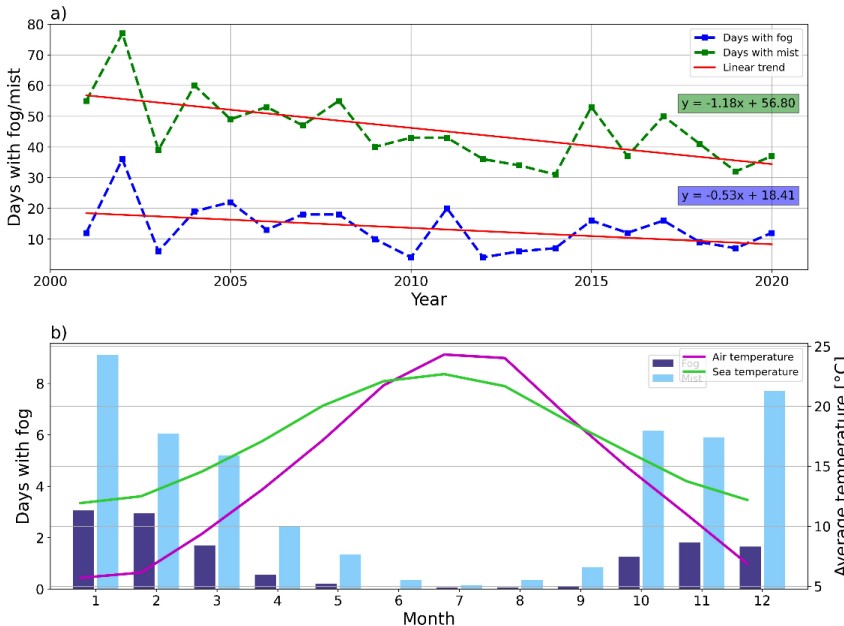

**Figure 2. a) Annual number of days with fog (blue line) and mist (green line) at Pula Airport, 2001-2020. with associated**
**linear trends and trend equations. b) Average monthly number of days with fog and mist at Pula Airport, the average sea**
**temperature measured at the oceanographic station in Pula Bay and the average air temperature measured at the Pula**
**Airport, 2021-2020.**
To obtain a complete picture of fog occurrence over Pula Airport, a climatological analysis was first performed on a
large scale and then on a smaller scale. Figure 2a shows the yearly number of days with fog and mist during the period
from 2001 to 2020. A day with fog is defined as any day with at least one METAR report indicating observed visibility
of less than 1 km and fog being reported at the airport. Similarly, a day with mist is defined as any day with at least
one METAR report indicating observed visibility of at least 1 km but less than 5 km (this is the aeronautical definition
of mist) and mist being reported at the airport. By adding the condition of fog or mist being reported, situations where
reduced visibility was due to precipitation—such as drizzle or rain—were excluded. On average, there are 13.4 days
of fog and 45.6 days of mist per year; however, this number is steadily decreasing, as indicated by the pronounced
negative trend. The Mann-Kendall statistical significance test shows that this result is statistically significant at the





95% confidence level. A careful evaluation of the linear trend reveals that the average number of fog days has
decreased by more than 10 days (from 18.4 in 2001 to 8.3 in 2020), and the average number of mist days has decreased
by more than 22 days (from 56.8 in 2001 to 34.4 in 2020). A more detailed analysis of fog and mist occurrence, this
time in relation to different seasons (Figure 2b), shows that the vast majority of fog and mist (more than 90%) at Pula
Airport occurs between October and March. During the observed multi-year period, January is the month with the
highest proportion of fog and mist days relative to the total number of days (23% and 20% respectively), followed by
February with 21% and 13% respectively, while December stands out only for mist with 17%. In March, October, and
November, the share of fog and mist days in the total number of days is 12-13%. Fog at Pula Airport can occur during
April and May, but from June to September, fog is an extremely rare phenomenon— for example, in June, there was
no recorded fog during the analyzed period from 2001 to 2020. Summarizing the data shows that during the
climatological summer (June-July-August), fog occurrence can be expected approximately every six years. The yearly
frequency of mist follows a similar pattern, with the only notable difference being that mist is a more frequent
phenomenon. The frequency of fog persistence, which is defined as fog occurrence on two consecutive days, is shown
in Figure 3. As expected based on previous findings about fog characteristics, persistence can be expected only in the
colder part of the year (October-April). From May to September, there have been no recorded instances of fog
occurring on two consecutive days. Stable anticyclonic conditions in the cold part of the year are most conducive to
fog persistence.

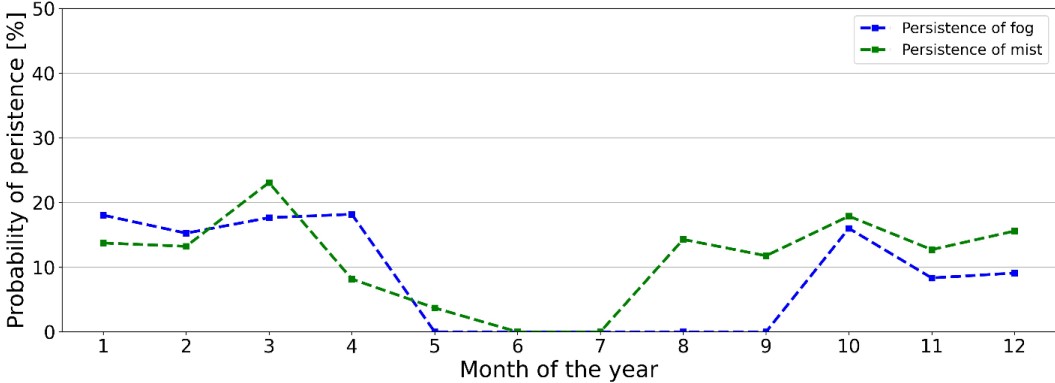


**Figure 3. Yearly distribution of the climatological probability of persistence (fog occurrence on two consecutive days) of fog**
**and mist at Pula Airport, 2001-2020. The probability of persistence is defined as the number of days with persistence relative**
**to the total number of days with fog in a given month. Cases where a single fog event was present around midnight (and**
**thus spanned two days) were not counted as persistence.**
Wind is one of the most important meteorological variables affecting the formation and maintenance of a fog layer,
as turbulence generated from wind shear greatly influences the height of the stable boundary layer. Therefore, it is
interesting to examine the statistical characteristics of wind during fog episodes. Figure 4 shows wind distributions
(data from METAR reports) for all data and for fog/mist conditions in a wind rose plot. In general, the dominant wind
at Pula is the NE bora wind, which can easily reach speeds greater than 10 m s$^{-1}$. Other common winds include
westerlies, a consequence of the superimposition of Etesian winds with the sea breeze circulation (Pandžić and Likso,
2005; Klaić et al., 2009), as well as north-northeasterly and southeasterly winds. Northwesterly and southwesterly



winds occur rarely at Pula Airport. The wind rose for fog conditions (Figure 4b) is markedly different; in these
situations, W/NW winds are the most common (47.7% of measurements). These winds blow from the direction of the
open sea, whereas easterly winds blow from Kvarner Bay, which is dotted with islands and where the sea is deeper.
Fog is also commonly encountered when there are light winds from the east, and there were some cases recorded with
winds from the north. Southerly winds are almost never associated with fog at Pula Airport. In the majority of cases
where fog formed in conditions with easterly winds, wind speed was lower than 3 m s$^{-1}$. During fog in westerly
conditions, wind speeds greater than 3 m s$^{-1}$ are far more common. When mist is reported (Figure 4c), the results are
reversed: mist is much more common in situations with easterly winds (28.8% of measurements), which are generally
the most frequently observed wind direction at Pula. This is unsurprising since mist is a more frequent phenomenon
than fog (Figure 2).

A closer look at the relationship between wind and visibility/cloud base in fog conditions (Figure 5a-d) provides
further evidence of the scarcity of fog in situations with northerly winds. The scatter plot of visibility and wind speed
confirms the existence of the aforementioned optimal window of wind speeds—the majority of fog forms when wind
speeds are between 0 and 2 m s$^{-1}$. The same is also observed for low clouds (lower than 200 m) — the majority of low
cloud bases were observed at wind speeds of 1 m s$^{-1}$ or less, some between 1 and 2 m s$^{-1}$, and very few at higher wind
speeds. The absence of cloud bases above 300 m at wind speeds higher than 2 m s$^{-1}$ is interesting. Higher wind speeds
indicate stronger advection, and personal communication from forecasters confirms that in these cases cloud bases
can be very low. This is particularly evident in situations with westerly flows, where fog is more often observed than
in easterly flows and where wind speeds are higher. The data for mist conditions (Figure 5e-h) leads to similar
conclusions for visibility—mist most often forms in westerly or easterly winds with somewhat higher wind speeds
than fog. Mist conditions with cloud bases above 500 m can occur but are rare, and one noteworthy difference between
mist and fog is the higher number of mist events with a low cloud base in southeasterly and southwesterly winds.

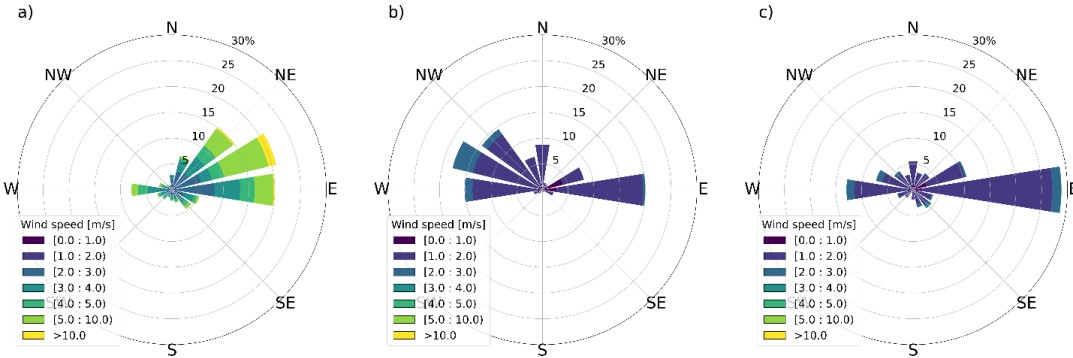

**Figure 4. Wind rose plots for Pula Airport, for dataset in the period 2001-2020: (a) the whole dataset, (b) fog conditions (c)**
**mist conditions. Data includes only reports where the variation in direction is less than 60° according to ICAO definition.**
**These account for 74 % of total data, 36 % of data in fog conditions and 39% of data in mist conditions.**



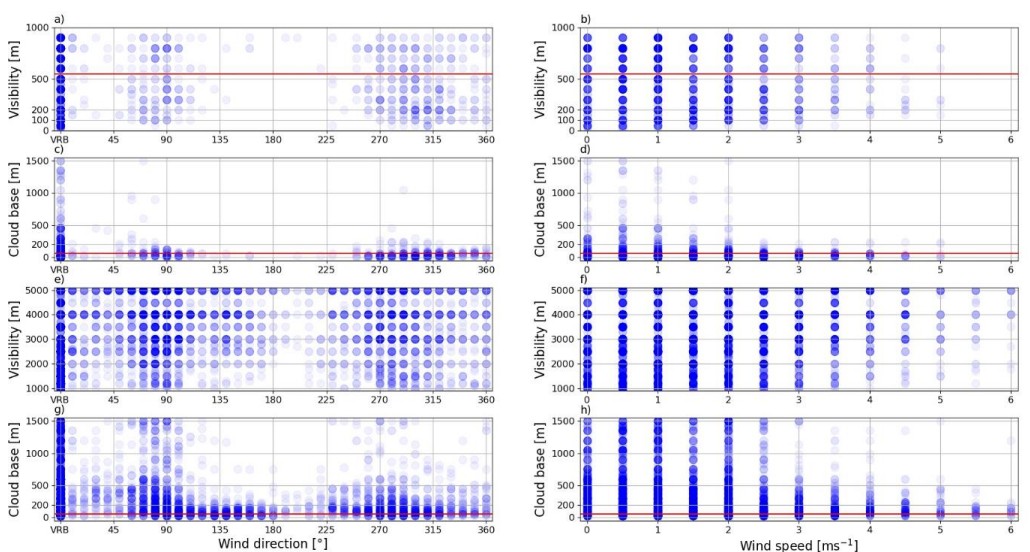

**Figure 5. Scatter plots of various meteorological parameters for fog conditions (a-d) and mist conditions (e-h) at Pula Airport, 2001-2020. Circles are colored according to the frequency of data points (darker – more frequent). Red lines mark the limits for Category I takeoff-landing procedures mentioned in Chapter 2.**

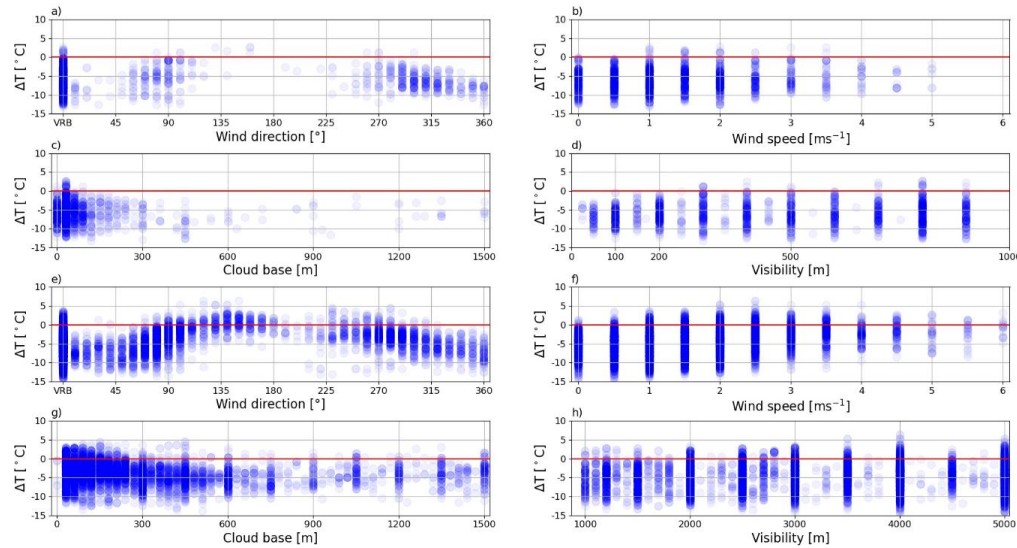

**Figure 6. Scatter plots of air-sea temperature difference between Pula Airport and Pula oceanographic station and various meteorological parameters at fog initiation (a-d) and mist initiation (e-h), 2001-2020. Circles are colored according to the frequency of data points (darker – more frequent). Red lines mark the 0 °C difference between air and sea temperature.**





Another way to examine fog occurrence is by analyzing the meteorological conditions prior to/at the time of fog
formation, as has been done in some previous studies (Tardif and Rasmussen, 2007; Veljović et al., 2015; Zoldoš and
Jurković, 2016), or by investigating the difference between air and sea surface temperature for marine fog (Li et al.,
2022). Since this study investigates fog at a coastal location, it would be of interest to investigate the difference
between air temperature (SAT) and SST (SAT-SST) during fog. For this purpose, SAT data from METAR reports at
Pula Airport were compared with SST data from the oceanographic station in Pula Bay, in the immediate vicinity of
the city of Pula. Figures 6a-d show scatterplots of the SAT-SST difference and various parameters at fog initiation
(first METAR report with visibility <1000 m), and the same is shown in Figure 6e-h for mist initiation. In general, the
majority of fog and mist cases form when there is a negative SAT-SST difference, i.e., when SST is higher than the
air temperature. Fog is a rare phenomenon during southerly winds (wind direction 135-225°), but mist in these
conditions is more common, especially in southeasterly winds (Figure 6e). During SE, S, and SW winds with mist
conditions, the SAT-SST difference is close to zero. In 97.4% of fog data, the SAT-SST difference was less than 0,
and in only 2.6% of data was it greater than or equal to zero. In mist conditions, there are slightly more cases where
the SAT-SST difference was greater than or equal to 0 (7.4%), but still the balance heavily favors warm sea-cold land
conditions.

Although the influence of SST was estimated by comparing the SAT values of the airport and the SST values of the
oceanographic station, a more comprehensive picture emerges when the influence of the sea to the west and east of
the airport is assessed. For this purpose, the satellite SST values were used for the area west and east of the airport
(Figure 1b), over which the two most prevailing winds at Pula Airport are common during fog formation. The SAT
and SST values were plotted in two scatter plots for the eastern and western areas for all observations, for observations
with fog, and observations with mist (Figure 7). A visual inspection shows that fog and mist usually occur when the
SAT-SST pairs have lower values. However, a closer look reveals that the data for fog (red dots) are more widely
distributed on the graph for the western region, indicating a greater scattering of observations for SST. This is
confirmed by calculations, which show that variance for SST of the western area (for observations with fog) is 12.62
°C, and for the eastern area, it is 11.37 °C. The same is true for observations with mist (16.24 and 14.66 °C,
respectively). Further insight into the influence of SST on fog formation was gained by separately analyzing the
differences between the SST of the western and eastern areas in cases with and without fog or mist (Figure 8). The
area west of Pula is generally warmer when neither fog nor mist is present, but fog occurs more frequently when the
SST of the eastern area is higher, while the opposite is true for mist. These results can be related to the wind rose
statistics for fog and mist (Figure 4), which shows that during fog, the prevailing wind is W-NW (promoting SST
decrease in the western area), while during mist, the prevailing wind is E (promoting SST decrease in the eastern area).
The temperature differences rarely exceed 0.5°C.



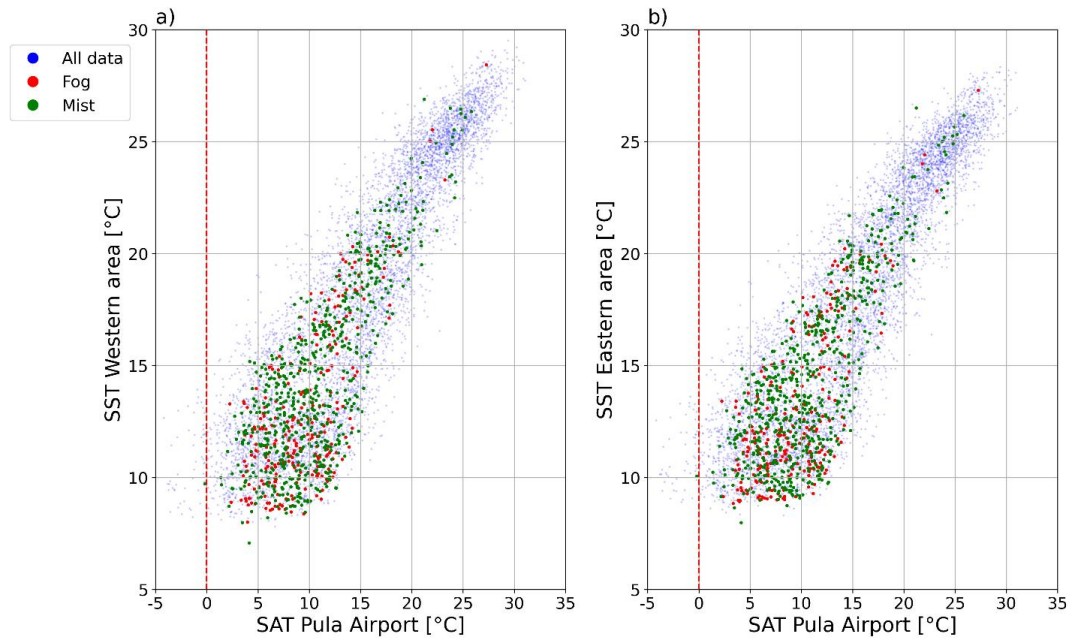

**Figure 7. Scatter plots of satellite sea surface temperature (SST) for western area (a) and eastern area (b), and surface air temperature (SAT) at Pula Airport, 2001.-2020.**

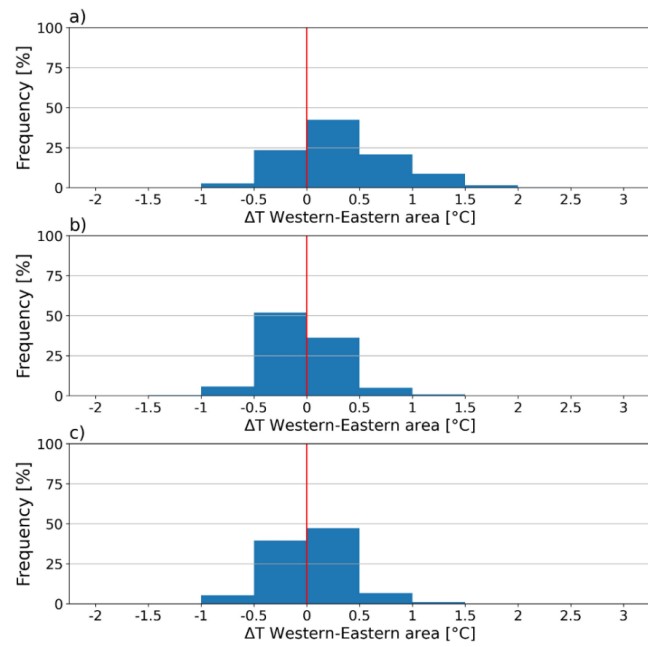

**Figure 8. Histograms of satellite SST difference between the western and eastern area for observations without fog or mist (a), fog (b) and mist(c).**





### 3.2 GNG analysis of synoptic weather patterns

In this analysis, the GNG method was applied to extract characteristic temporal and spatial patterns of wind and MSLP fields that favor the formation of fog and mist in the Pula region. The GNG analysis was performed on a monthly basis for the entire period analyzed; i.e., all January data, February data, etc., were entered separately in order to calculate them. The reason for this approach is that it makes the results easier to interpret, captures seasonality, and reduces the computational load, which is important given the large hourly dataset involved. To save computational resources, the GNG algorithm was applied only to 10-m wind data from ERA5, and the derived pressure fields were subsequently extrapolated. To obtain an appropriate synoptic situation with large wind systems, such as anticyclones and cyclones affecting the northern Adriatic, a large area in the Mediterranean was selected for analysis (Figure 1b). The 40-year period of ERA5 data (from 1979 to 2019) provided long time series data from which the GNG analysis could learn, enabling the derivation of more accurate and robust spatio-temporal patterns. The one-year discrepancy in the overlap of the ERA5 and fog datasets, particularly in 2020, is a consequence of the fact that the GNG analysis was performed prior to the start of this paper and the computational resources required for a new analysis were no longer available. Given that wind was the primary variable and MSLP the derived variable, wind exerted the predominant influence, thus excluding the occurrence of very strong cyclonic (MSLP < 1000 hPa) or anticyclonic (MSLP > 1030 hPa) formations in the results. However, from a conceptual point of view, and with regard to the dynamic interaction between high- and low-pressure systems, the results are satisfactory and align well with the climatology of the region. To facilitate the observation of broader wind patterns, only wind fields over the sea were visualized. This choice was made because wind patterns over the sea are more uniform and consistent than wind patterns over land, where local variations such as topography or land features can have a large influence on the results.

This process resulted in 9 Best Matching Units (BMUs) for each month (a total of 108 BMUs) distributed over hourly data, representing the weather patterns with the highest data variance. The hourly data were aggregated into daily data by identifying the most frequent BMU within a day. Subsequently, each day with fog/mist in the 2001-2019 period was associated with the dominant BMU for that day, resulting in a synoptic weather pattern classification for fog/mist days at Pula Airport. To focus on the prevailing synoptic patterns that contribute to the formation of fog and mist, months with more than 10% of days with fog and mist were arbitrarily selected (Table 1). These months are in the colder part of the year and together account for more than 85% of the foggy days of the year. A criterion relating to BMUs was also arbitrarily defined for each month: Only BMUs with more than 15% representation in a given month were selected (Table 1) and plotted (Figures 9-14). In this way, 2 to 4 BMUs were selected and more closely analyzed for each month.





**Table 1. Display of the most frequent Best Matching Units (BMUs) (whose monthly share is greater than 15 percent) describing the prevailing synoptic weather pattern during the days with fog and mist for the selected month at Pula Airport, 2001-2019. Blue colored denotes BMU common (>15 % share) for fog and mist, green colored for fog only and red colored for mist only. The slope coefficients describe the linear trends of the most common BMUs, i.e. the yearly change in frequency.**

| Month | BMU | Synoptic pattern | Wind | Slope coeff. | Frequency (fog) | | Frequency (mist) | |
|---|---|---|---|---|---|---|---|---|
| | | | | | # | % | # | % |
| January | BMU-1-6 | Quasi-non-gradient-field | WNW, W | -0.390 | 30 | 54 % | 68 | 39 % |
| | BMU-1-8 | Cyclone over northern Adriatic (MSLP<1008 hPa) | WNW, NW | 0.107 | 11 | 20 % | 39 | 23 % |
| February | BMU-2-5 | Anticyclone over central/western Europe (MSLP>1028 hPa) | NE, NNE | 0.006 | 8 | 16 % | 13 | 12 % |
| | BMU-2-6 | Quasi-non-gradient-field | W, SW | -0.138 | 17 | 33 % | 36 | 32 % |
| March | BMU-3-1 | Anticyclone over southeastern Europe (MSLP>1022 hPa) | S, SE | -0.572 | 16 | 16 % | 5 | 16 % |
| | BMU-3-3 | Cyclone over northern Adriatic (MSLP<1006 hPa) | SE | -0.051 | 6 | 19 % | 25 | 25 % |
| | BMU-3-8 | Quasi-non-gradient-field | S, SE | 0.255 | 11 | 35 % | 31 | 39 % |
| October | BMU-10-1 | Anticyclone over southeastern Europe (MSLP>1018 hPa) | NE | 0.158 | 4 | 16 % | 13 | 11 % |
| | BMU-10-5 | Anticyclone over southeastern Europe (MSLP>1020 hPa) | SE | -0.272 | 10 | 40 % | 38 | 32 % |
| | BMU-10-6 | Quasi-non-gradient-field | W | -0.051 | 4 | 16 % | 27 | 23 % |
| November | BMU-11-5 | Anticyclone over eastern Europe (MSLP>1024 hPa) | SE | 0.251 | 8 | 24 % | 23 | 21 % |
| | BMU-11-7 | Quasi-non-gradient-field (MSLP>1022 hPa) | WSW, W | -0.415 | 6 | 18 % | 29 | 26 % |
| | BMU-11-9 | Anticyclone over southeastern Europe (MSLP>1026hPa) | NE | -0.041 | 11 | 33 % | 28 | 25 % |
| December | BMU-12-1 | Anticyclone over southeastern Europe (MSLP>1026 hPa) | NE | -0.000 | 7 | 21 % | 31 | 21 % |
| | BMU-12-3 | Cyclone over southern Adriatic (MSLP<1008 hPa) | NE | 0.266 | 5 | 15 % | 23 | 15 % |
| | BMU-12-4 | Cyclone over the Tyrhennian Sea (MSLP<1008 hPa) | NE | -0.470 | 6 | 18 % | 14 | 9 % |
| | BMU-12-8 | Anticyclone over central and eastern Europe (MSLP >1030 hPa) | NE | -0.130 | 3 | 9 % | 23 | 15 % |

352

The analysis of synoptic patterns favoring fog formation (Table 1) shows that in January, the prevailing synoptic conditions favoring fog and mist in Pula are characterized by the dominance of a quasi-non-gradient field over the entire region (BMU-1-6, Figure 9a), where the mean pressure gradient is very weak (Belušić Vozila et al., 2021). This stable atmospheric pattern supports the maintenance of a relatively calm and stagnant air mass over Pula, which inhibits the dispersion of pollutants and moisture, promoting fog formation in the area. The second most common synoptic pattern that favors the formation of fog and mist is the one with a cyclone over the northern Adriatic (BMU-1-8, Figure 9b). Both synoptic patterns support weak WNW/NW winds over the Istrian peninsula. A similar synoptic pattern can be observed during the transition to the months of February and March. Again, the quasi-non-gradient field is the most common pattern suitable for the formation of fog and mist in February (BMU-2-6, Figure 9d) and



March (BMU-3-8, Figure 10c), but compared to January, weak W and SW winds are also present in February, and
weak S and SE winds are present in March. In addition to the quasi-non-gradient field, the favorable synoptic patterns
in February and March also include anticyclones in the continental part of Europe (BMU-2-5, Figure 9c and BMU-3-
1, Figure 10a) and cyclones in the northern Adriatic (BMU-3-3, Figure 10b). Few fog events occur in Pula from April
to September, so the BMU results cannot reveal any clear patterns. The limited occurrence of fog during this period
is attributed to the lower relative humidity and more stable atmospheric conditions typical for summer in this area.
Nevertheless, local factors such as the sea/land breeze and coastal geography may play a role in the occurrence of fog.
Consequently, any analysis based on these months should be treated with caution, as the number of recorded fog
events is low.

At the beginning of October, the synoptic conditions that favor the occurrence of fog and mist in Pula are typically
characterized by an area of high pressure over continental Europe, with the most common position being where the
center of the high-pressure area is over southeastern Europe, supporting SE winds (BMU-10-5, Figure 11b). This
anticyclonic trigger persists into November, accompanied by a strengthened mean pressure field and intensified
synoptic pressure gradients over the central Mediterranean. These conditions favor the development of fog in the
region (BMU-11-9, Figure 12c). The intensified synoptic pressure gradients over the central Mediterranean contribute
to increased NE wind patterns. This increased wind activity can result in moist air being transported from the sea to
the coastal regions, providing an additional source of moisture for fog formation. The convergence of air masses along
these enhanced pressure gradients can also promote upward movements, leading to local cooling and increased
humidity. Pula's coastal location further reinforces the influence of the anticyclone. Coastal areas are more susceptible
to temperature inversions due to their proximity to the sea, which retains heat and reduces temperature fluctuations.
Anticyclonic conditions over Eastern Europe combined with coastal geography create an environment where cool,
moist air is trapped near the surface, favoring the formation of fog. The conditions that favor mist formation are
somewhat varied; the most common synoptic pattern is the one where the high-pressure area is located over the
western Mediterranean and WSW winds blow over the Istrian peninsula. In addition to the anticyclone, a quasi-non-
gradient field with weak W winds also occurs in October and November (BMU-10-6, Figure 11c and BMU-11-7,
Figure 12b). In December, the prevailing synoptic weather pattern associated with fog and mist becomes more difficult
to recognize – the frequency of the most common BMUs is more evenly distributed, and in addition, more dynamic
synoptic conditions prevail (similar to February and March). Nevertheless, the most frequent synoptic pattern for fog
and mist (BMU-12-1, Figure 13a and BMU-12-8, Figure 13d) has anticyclonic dominant features. Under these
conditions, the Pula region is under the influence of a pressure ridge, with the prevailing weak wind patterns from the
northeast further increasing the probability of fog formation. Compared to the previous months, there is also a
pronounced influence of cyclones on the formation of fog and mist in December, in two respects: on the one hand, by
a weak SE wind due to a cyclone in the southern Adriatic (BMU-12-3, Figure 13b) and on the other hand by a stronger
NE wind due to a strong cyclone in the Tyrrhenian Sea (BMU-12-4, Figure 13d).


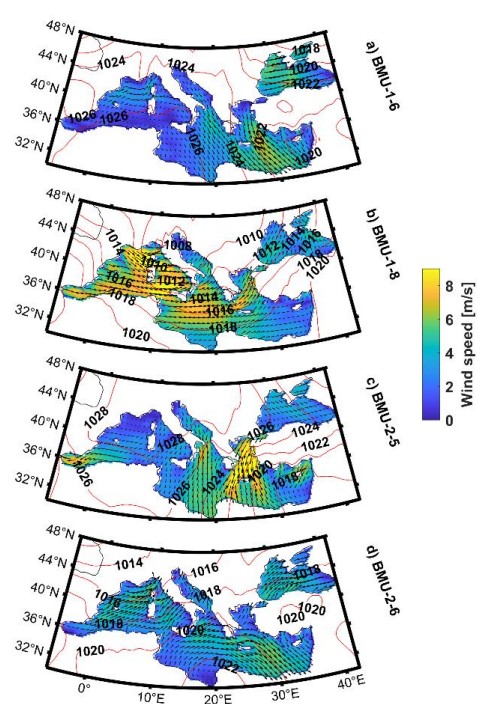

**Figure 9. Most prevailing best Matching Units (BMUs) describing synoptic patterns favoring the formation of fog and**
**mist at Pula airport in January (a-b), February (c-d).**

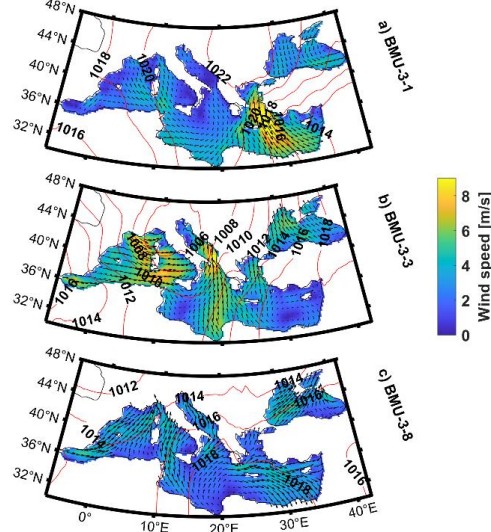

**Figure 10. Most prevailing best Matching Units (BMUs) describing synoptic patterns favoring the formation of fog and**
**mist at Pula airport in March (a-c).**


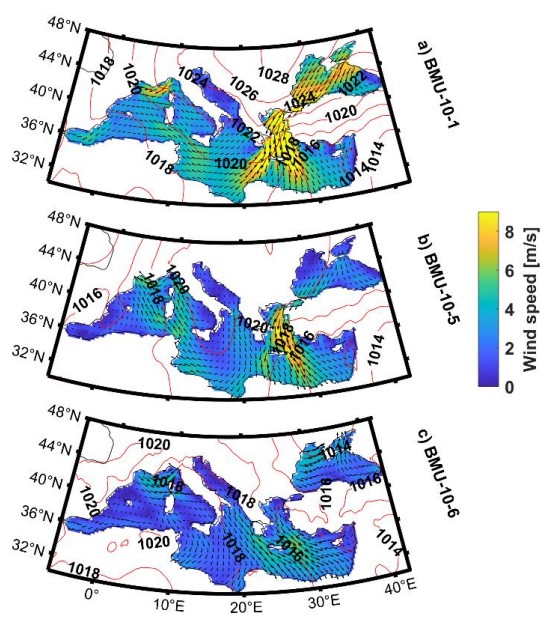

**Figure 11. Most prevailing best Matching Units (BMUs) describing synoptic patterns favoring the formation of fog and**
**mist at Pula airport in October (a-c).**

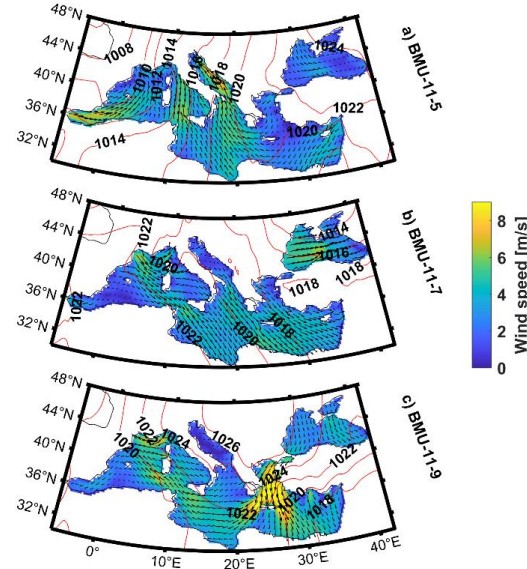

**Figure 12. Most prevailing best Matching Units (BMUs) describing synoptic patterns favoring the formation of fog and**
**mist at Pula airport in November (a-c).**

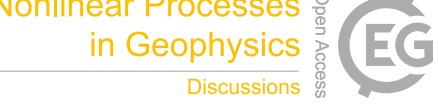

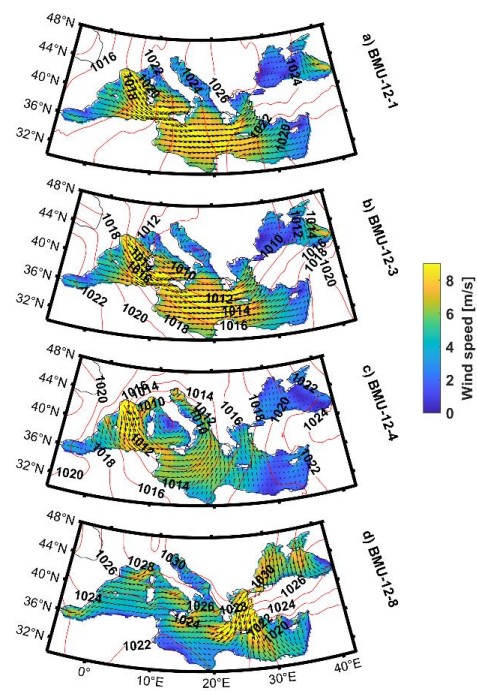

**Figure 13. Most prevailing best Matching Units (BMUs) describing synoptic patterns favoring the formation of fog and**
**mist at Pula airport in December (a-d).**

In addition to observing the prevailing synoptic patterns during the occurrence of fog and mist, the GNG analysis also
enables the investigation of the time series of frequencies of individual BMUs. This was done by first calculating the
relative monthly frequency of each BMU for each year, and then using linear regression to estimate a trend and
calculate slope coefficients. This ultimately offers insight into changes in the frequency of each BMU throughout two
decades (Table 1, Figure 14). In January, it can be seen that the more common BMU-1-6 shows a strong negative
trend, while the less frequent BMU-1-8 shows a weaker positive trend. In February, the most common BMU (BMU-
2-6) shows a weak negative trend. In March, the most common BMU (BMU-3-8) shows a positive trend, while the
frequency of BMU-3-1 shows a strong negative trend. In October, there is a decrease in the most common BMU for
fog and mist (BMU-10-5), while November and December show major changes in the frequency of less common
BMUs. Analyzing the data by counting the positive and negative changes and dividing the synoptic patterns into three
groups (cyclonic/anticyclonic/quasi-non-gradient field) shows that of the 17 BMUs, 8 are anticyclonic, 5 are quasi-
non-gradient field, and 4 are cyclonic. There is no trend for 1 BMU (BMU-12-1) in the anticyclonic patterns; 3 BMUs
increase in frequency and 4 BMUs decrease in frequency over the years. For quasi-non-gradient, there is 1 increase
and 4 decreases. Of the 4 cyclonic BMUs, 2 increase in frequency and 2 decrease. In addition, a strong decrease (slope
coefficient >0.3 or <-0.3) is observed in BMU-3-1 (anticyclonic), BMU-1-6 (quasi-non-gradient field), and BMU-12-
4 (cyclonic). Positive trends do not have such large values for the gradient coefficient.

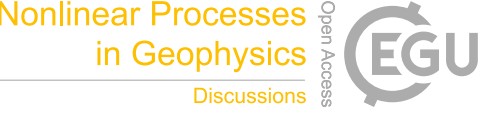

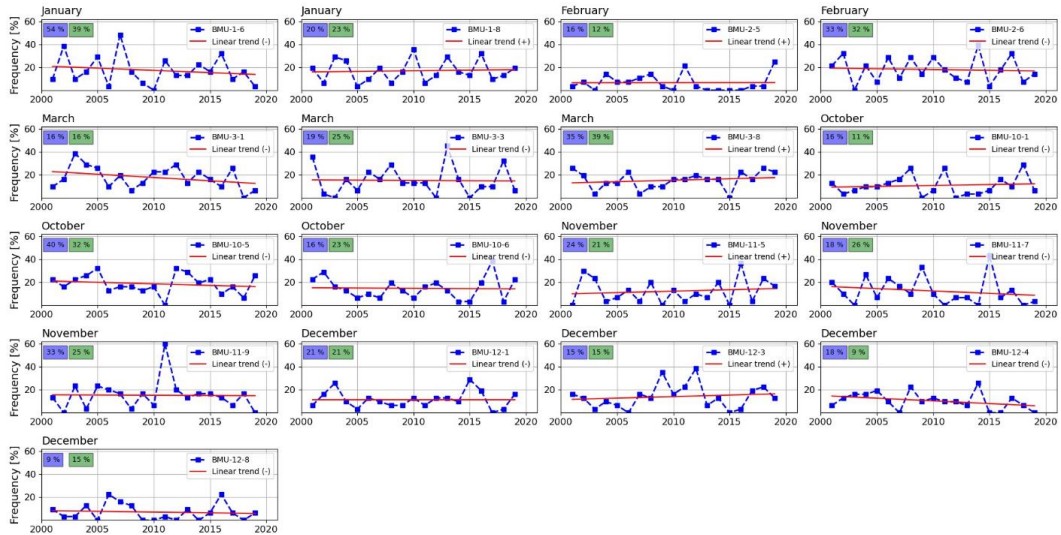

**Figure 14. Relative frequencies and trends of most common (monthly share in days with fog/mist greater than 15%) BMUs for fog and mist in Pula Airport, data period: 2001-2019. Numbers shaded in blue denote the share of a BMU for fog days, numbers shaded in green denote the share of a BMU for mist days.**

## 4 Discussion and conclusion

This comprehensive climatological analysis of fog and mist occurrences at Pula Airport from 2001 to 2020 has provided valuable insights into the changing patterns of these meteorological phenomena. Given that the occurrence of fog in the eastern Adriatic has been rarely studied, with the last studies conducted 50 years ago (Stipaničić, 1972), there was a strong need for new insights. In summary, instead of the prevailing purely statistical approach to fog and mist occurrence, a dual approach was chosen that combines classical statistics with neural networks, integrating measured airport data with synoptic parameters that favor fog and mist formation. This was particularly important, as the airport's meteorologists already had operational experience regarding which weather patterns, in conjunction with SST, favor fog formation, but this knowledge had not yet been scientifically published.

One of the first important results is the statistically significant decreasing trend in the frequency of fog and mist in the region, consistent with similar observations in Europe, such as at Zagreb Airport (Zoldoš and Jurković, 2016) or Milano Airport (Mariani, 2009). While the decreasing trend in Zagreb and Milan can be largely attributed to a reduction in air pollutants, this conclusion is more difficult to draw for Pula, as it is a much smaller city with less developed industry, whose impact on the neighboring suburban/rural areas is not as great. The decrease in fog and mist frequency has been observed in Europe, but the effect is most pronounced in continental Europe and not so much in the Mediterranean region (Vautard et al., 2009). Apart from the common negative trend, the annual distribution of fog at Pula Airport, where 90% of events occur from October to March, is also comparable to the annual distribution of fog events at Zagreb Airport (Zoldoš and Jurković, 2016).



The second important climatological finding is that, in recent decades, the maximum occurrence of fog in the winter
months has shifted from February to January. While February used to be the foggiest month at Pula Airport, with an
average of 3.5 foggy days (Stipaničić, 1972), the most recent data show that January has become the foggiest month
with 3.05 foggy days (Figure 2b). February remains the second foggiest month, with an average of 2.95 foggy days.
This result underscores the importance and value of long data series, which can provide crucial information such as
this shift in fog frequency.

The third climatological conclusion is that fog is most likely to occur with westerly and northwesterly winds (47.7%
of measurements), while mist is most common with weak easterly winds. Additionally, compared to continental
locations such as Zagreb Airport (Zoldoš and Jurković, 2016), more fog occurs at Pula Airport at wind speeds
exceeding 3 m s$^{-1}$. Conversely, fog at moderate or strong winds (> 3 m s$^{-1}$) is less common at Pula Airport than in
locations bordering the open ocean, such as the coast of California, where fog often forms at wind speeds above 10 m
s$^{-1}$ (Filonczuk et al., 1995). Another finding is that fog rarely forms in calm or very low wind conditions, suggesting
that there may be an optimal window of wind speeds that favor fog formation; this part remains open for future
investigation. Gultepe et al. (2007) emphasized the importance of wind speed and its influence on turbulence and heat
fluxes at the surface, which in turn significantly affect fog occurrence. It is important to note that these results are only
based on data where the wind direction is not variable (VRB), i.e., where the change in direction is less than 60°, as
defined by the ICAO (International Civil Aviation Organization). The total proportion of non-VRB data is 74%
overall, 36% for fog conditions, and 39% for mist conditions.

As a second approach to gaining insights into fog and mist events at Pula Airport, the Growing Neural Gas (GNG)
method was used to analyze the synoptic patterns (BMUs) that favor fog formation at Pula Airport. Fog and mist
occurrences are most pronounced in the colder part of the year, but the GNG allowed for a deeper analysis. The
majority of the fog events in the cold season occur during stable anticyclonic or quasi-non-gradient field conditions
(13 in total), although some events also occur during low-pressure conditions (4 in total). In January, February, and
March, anticyclonic and quasi-non-gradient fields were the most frequent patterns, while the cyclone field was the
second most frequent in January and March. In October and November, anticyclonic and quasi-non-gradient fields
predominated uniformly, contrasting with December, where anticyclonic and cyclonic patterns coincided. These
machine learning results are consistent with the empirical knowledge of airport forecasters (personal communication),
who concluded that two weather types are most frequently associated with fog in Pula. The first is a predominantly
westerly flow bringing in moist air from the west (W) and northwest (NW) during anticyclonic conditions. The second
weather type is a predominantly easterly flow with advection from the southeast (SE) during a weakening anticyclone.
January and February are precisely the months in which the most frequent pattern is a quasi-non-gradient field
characterized by weak westerly winds in the Istrian area, and March is the month in which the SE wind dominates in
more than 60% of cases. An interesting exception is December, which deviates from this rule and is dominated by NE
winds. Combining the two approaches by comparing the GNG results with the wind rose, it can be seen that the
dominance of W/NW winds (more than 60% of cases) significantly contributes to January being the month with the





most fog and mist days (Figure 4, Table 1). Considering the total number of fog and mist days in January, it is not
surprising that this wind type can have a large share in the overall annual variability of winds favorable for fog.
However, the third approach, which involves the use of satellite SST data, must also be considered. If we take into
account W/NW winds, which advect air at low speed over the warm sea west of the coast of Istria, the result is clear:
warm, humid air is transported over the cold surface around the airport, supporting the formation of fog and mist.
Another conclusion, which again refers to the GNG results, the wind rose, and the SST, points to December. In
December, fog is caused by weak northeasterly winds, which can result from both low- and high-pressure fields.
However, these winds advect air from the east over the enclosed sea of the Kvarner Gulf, which is still warm. This
means that warm air still exists above the warm sea surface, which can be advected to the west. Therefore, in January,
when the sea in Kvarner has already cooled down significantly, the fog at the airport can be advected from the warmer
open sea, which has not yet cooled down sufficiently. In both cases, the SST for both the western and eastern areas is
2 to 6 degrees greater than the SAT at the airport.

Global warming and climate change are the most apparent explanations for the long-term decrease in fog frequency.
There are well-documented trends indicating rising temperatures in Pula and surrounding coastal and interior areas of
Istria (Bonacci, 2010; Šimunić et al., 2021), increased sea surface temperatures (SST) across the Mediterranean (Pastor
et al., 2018) and specifically in Pula (Bonacci, 2023), and growing ocean stratification worldwide over the past 50
years (Li et al., 2020). Climatological model reanalyses for the Adriatic region from 1987 to 2017 show positive SST
trends for every month of the year. Summer records the most significant increase, although winter months are also
experiencing notable rises (Tojčić, 2023). In terms of air temperature, reanalysis indicates that trends are more
pronounced from April to September compared to October to March (Tojčić, 2023; Bonacci et al., 2021). Regarding
advection fog and mist, wind trends must also be considered. Positive wind trends have been observed over the sea
and along the Adriatic coast, ranging between 0.1 and 0.2 m s$^{-1}$ per decade (Tojčić, 2023). Although these trends are
not large enough to overlap the optimal wind speed window for fog, they can significantly alter fog formation
mechanisms. For example, the decrease in temperature difference between the sea surface and air (due to rising SST)
reduces the temperature gradient, making it harder for air to cool to the dew point required for fog formation.
Additionally, higher SSTs result in increased evaporation rates, which, combined with favorable winds, can lead to
fog advection. In addition to local factors such as SST, air temperature and wind, changes in synoptic patterns that
favor fog or mist formation can be linked to broader synoptic trends over this part of Europe. A decrease in quasi-non-
gradient synoptic situations in January, February, October, and November has been documented in the summer months
(Belušić Vozila et al., 2021). One significant negative gradient is observed in the BMU cyclone pattern in December,
associated with its northward shift (Reyers et al., 2016; Knippertz et al., 2000) and noted in future climate research
for this region (Belušić Vozila et al., 2021). However, examining all BMUs with negative trends reveals a total of 10,
compared to 6 positive BMUs, and these negative BMUs show a higher frequency of foggy days per month (Table 1).
For example, BMU-1-6 has a frequency of 54% in January, BMU-2-6 is 33% in February, and BMU-12-4 is 18% in
December. This suggests that the reduction in favorable synoptic situations for fog and mist leads to a decrease in the
number of foggy days throughout the study period, outweighing other factors like positive SST trends. Although the

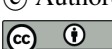



similarity to general synoptic system trends is notable, the focus of this work was not on the most common synoptic
patterns overall but rather on those showing the highest number of fog and mist days.

Overall, these results lay a strong foundation for further research and analysis, enabling a deeper understanding of the
various meteorological and oceanographic factors that influence fog and mist occurrence at Pula Airport. This is
particularly significant as it represents the first scientific study on fog in the Pula area in 50 years, a period during
which climate changes have notably altered the local climate. Future projections suggest these changes will become
even more pronounced, including lower wind speeds in coastal areas and more extreme contrasts such as increased
droughts and heavy precipitation events (Tojčić, 2024). This study has undertaken the broad task of identifying
synoptic patterns conducive to fog and mist formation. Since fog and mist formation is primarily influenced by wind
speed and moisture advection, there is potential for atmospheric-oceanographic coupled modeling that incorporates
finer local topography and improved parameterization of processes at smaller temporal and spatial scales. Such
advancements would contribute to a more comprehensive understanding of local meteorological phenomena and their
implications for various applications, including aviation meteorology and environmental monitoring.
**Code/data availability**
All data and codes used in the analysis are available from the corresponding author on request.
**Author contribution**
Study conception and design, material collection, data preparation and analysis were performed by Marko Zoldoš and
Tomislav Džoić. The manuscript was written by Marko Zoldoš, with contribution from Tomislav Džoić in the
"Discussion and Conclusion" section. All authors read and approved the final manuscript.
**Competing interests**
The authors declare that they have no conflict of interest.
**Acknowledgments**
Frano Matić was supported in part by the European University of the Seas (SEA-EU) alliance through collaborative
efforts and resources. Measurements and observations for Pula Airport were provided by the Croatian Air Navigation
Service (Crocontrol Ltd.). SST data for Pula were provided by the Meteorological and Hydrological Service of Croatia
(DHMZ). SST data in the Mediterranean region were downloaded from the Copernicus Marine Data Store
(https://data.marine.copernicus.eu/). 10-m wind and mean sea level pressure (MSLP) data were adopted from the fifth
generation of ECMWF's ERA5 reanalysis of global climate and weather. Final proofreading (grammar/spelling check)
was performed by ChatGPT from OpenAI.





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
