# Peer review of "Statistical and neural network assessment of the climatology of"

_Nonlinear Processes in Geophysics, 2024_

## Author Response (AR1)

Zagreb, 12 January 2025

Editorial Team

Nonlinear Processes in Geophysics

Dear Editor,

Please find enclosed a revised version of the manuscript titled "*Statistical and neural network assessment of the climatology of fog and mist at Pula airport in Croatia*" (the title was changed according to the proposal from Reviewer 2), revised according to the reviewers' suggestions.

We would like to thank the reviewers for their constructive suggestions and comments. We greatly appreciate the time and effort they dedicated to the review process. We hope that this new version of the manuscript is now more clearly presented and interpreted.

We have provided two versions of the manuscript: one with colored text corresponding to the reviewers' comments (Reviewer 1 – yellow, Reviewer 2 – blue), and the other with normal text. The text was color-coded to facilitate tracking changes made to the manuscript, and additionally all the major changes are also referenced in this document. Given the extensive modifications (including major revisions and grammatical corrections throughout the entire text), using the traditional track changes feature in Word would have resulted in an overwhelming number of edits, making it challenging and time-consuming for reviewers to follow all the updates. To address this, we opted for this alternative approach.

Below, we present the reviewers' comments in regular text, while our responses are highlighted in yellow (Reviewer 1) and blue (Reviewer 2).

On behalf of myself and all co-authors,

Yours sincerely,

Marko Zoldoš

**Reviewer #1:**

**Comment 1.** Line 143: Remove the '24'

Reply: The typo was corrected.

**Comment 2.** Figure 2b: The legends are overlapped.

Reply: The figure was improved (also according to suggestions from the other reviewer) and the overlapping corrected.

**Comment 3.** Line 212: 'Summarizing the data should be 'A summary of the data'

Reply: The phrasing was revised accordingly.

**Comment 4.** Figure 5: Explain what 'VRB' is in the caption.

Reply: The explanation was added to captions of figures 5 and 6:

*"VRB" denotes variable wind direction (180° or more in a 2-minute interval) according to definition by ICAO (International Civil Aviation Organization).*

**Comment 5.** Figure 9: Use a dot to indicate where the airport is. The coastal line is not clear. Should use bold line. In the caption, say that you only plot the wind vectors and shadings for the sea - indicate the contour lines are the MSP. The wind vectors are not visible near the airport. It is better to plot additional panels (e, f, g, h) focusing on the region near the airport. This smaller region can be the one in Figure 1b. Same for figure 10-13.

Reply: The figure 9 was improved: the coastline was made bolder and the airport was clearly marked with a dot. Additional panels were made for the region near the airport. The required convention was slightly changed compared to the reviewer's requirement: the additional fields in the second column have been labeled b, d, f, h and those in the first column a, c, e, g, because I've seen this in most papers. This region covers the central and northern Adriatic, so the wind vectors near the airport are easy to spot and the overall regime is immediately recognizable. Changes have also been made to the main text - in the section describing the results and in the discussion and conclusions. A smaller region as in Figure 1b could not be used because of two reasons: 1. the coarse horizontal ERA5 resolution of 0.25° (only a few vectors would be visible), and 2. the wind vectors over land were never used since the main idea of this work was to illuminate the synoptics and the winds over the sea. Over the sea, the winds are very homogeneous and are often directly related to large-scale synoptic systems such as cyclones or anticyclones, whereas over land, the influence of topography and vegetation introduces noise, masking these processes.

**Comment 6.** Figure 10-13. The captions can be simplified to 'similar to Figure 9 but in March', and so on.

Reply: The captions have been simplified, example:

*Figure 10. Similar to Figure 9, but for February (a) and March (c, e, g) in the wider Mediterranean area, and for the northern and central Adriatic: February (b) and March (d, f, h) in the zoomed-in area.*

**Comment 7.** Line 433-540: The Discussion and summary section is too long. It is difficult to understand the main point. Many contents should be integrated into section 3.

Reply: The "Discussion" and "Conclusion" sections have been revised in line with the reviewer's suggestions. Most of these sections have been merged with section 3. The text has been simplified in order to present the most important results of this work clearly and concisely.

**Reviewer #2:**

**Comment 1.** Using simple statistics and a more sophisticated artificial neural network approach, this study aims at describing synoptic conditions leading to the formation of fog and mist in the vicinity of Pula airport (Croatia). It is an interesting research topic, knowing the negative impact of fog/mist on the airport operations.

However, there are several problems, listed below, that need to be addressed before proceeding further. Also, the text is sometimes very scholar. It would be very beneficial for this work to have a more scientific writing.

I therefore recommend major revisions.

Reply: We thank the reviewer for all the comments and instructions. The paper was revised according to the reviewer's instructions. The manuscript was rewritten as scientifically as possible, especially the conclusion, which was completely revised with the aim of conciseness and clarity.

**Comment 2.1** The "Discussion and conclusion" part is too long and often difficult to follow. Most of the content is just a repetition of what has been presented in the previous section, and should be merged with section 3. According to the journal guidelines (https://www.nonlinear-processes-in-geophysics.net/submission.html#manuscriptcomposition), the last section of the main text should only present the conclusions.

Reply: The "Discussion" and "Conclusion" sections have been revised in line with the reviewer's suggestions and journal guidelines. The Conclusion is now shorter, more concise and clearer in its presentation of the conclusions of this paper. Excessively long descriptions and repetitive content have been moved to section 3.

Several parts need to be rewritten to improve the quality of the manuscript:

L.296-298: What is the meaning of these values?

Reply: The relevant sections have been revised accordingly and the text has been changed so that it is now easier to read. In L296-298, the variance has been replaced by the standard deviation in order to show the dispersion more clearly.

Old version of text:

*However, a closer look reveals that the data for fog (red dots) are more widely distributed on the graph for the western region, indicating a greater scattering of observations for SST. This is confirmed by calculations, which show that variance for SST of the western area (for observations with fog) is 12.62 °C, and for the eastern area, it is 11.37 °C. The same is true for observations with mist (16.24 and 14.66 °C, respectively).*

Was replaced with this:

*A more detailed analysis reveals that fog data points (red dots) in the western region are more dispersed, indicating greater variability in SST for the same SAT, while the eastern region shows a more consistent SST-SAT relationship. Statistical analysis supports this observation, with the standard deviation of SST for fog observations being 3.55 °C in the western area and 3.37 °C in the eastern area. For mist, the standard deviation is 4.03 °C in the western area and 3.83 °C in the eastern area.*

L.298-304: I could not follow this part. Please clarify. –

Reply: The parts in question were revised and text was clarified.

Old version of text:

*The area west of Pula is generally warmer when neither fog nor mist is present, but fog occurs more frequently when the SST of the eastern area is higher, while the opposite is true for mist. These results can be related to the wind rose statistics for fog and mist (Figure 4), which shows that during fog, the prevailing wind is W-NW (promoting SST decrease in the western area), while during mist, the prevailing wind is E (promoting SST decrease in the eastern area). The temperature differences rarely exceed 0.5°C.*

Was replaced with this:

*To further investigate the role of SST in fog formation in fog formation, differences in SST between the western and eastern regions were analyzed under various conditions (Figure 8). When neither fog nor mist is present, SST in the western area is slightly higher than in the eastern area. During fog events, however, the pattern reverses, with SST being higher in the eastern area. In contrast, mist typically occurs when SST is higher in the western area than in the eastern area. These differences align with the prevailing wind directions observed during fog and mist events (Figure*

*4). During fog, winds that are predominantly from the west to northwest (W-NW) promote cooling in the western area. Winds from the east (E) during mist contribute to reduced SST in the eastern region. Despite these variations, the SST differences between the two areas are small, rarely exceeding 0.5 °C.*

L.413-428: Please rewrite to include information on linear trends and implication for the fog/mist. –

Reply: The parts in question were rewritten to include information on linear trends.

Old version of text:

*In January, it can be seen that the more common BMU-1-6 shows a strong negative trend, while the less frequent BMU-1-8 shows a weaker positive trend. In February, the most common BMU (BMU-2-6) shows a weak negative trend. In March, the most common BMU (BMU-3-8) shows a positive trend, while the frequency of BMU-3-1 shows a strong negative trend. In October, there is a decrease in the most common BMU for fog and mist (BMU-10-5), while November and December show major changes in the frequency of less common BMUs.*

Was replaced with this:

*In January, the more frequent BMU-1-6 (Figure 14a) has a strong negative trend of -0.390, while the less frequent BMU-1-8 (Figure 14b) has a weaker positive trend of 0.107. In February, the most frequent BMU-2-7 (Figure 14e) has a weak negative trend of -0.025. In March, the most frequent BMU, BMU-3-8 (Figure 14h) has a positive trend of 0.255, while BMU-3-1 (Figure 14f) has a strong negative trend of -0.572. In October, the most frequent BMU for fog and mist (BMU-10-5, Figure 14j) has a negative trend (-0.272), while in November and December there is almost no trend for the most frequent BMUs (BMU-11-9, Figure 14n and BMU-12-1, Figure 14o). It can be summarized from the table that the BMUs most associated with the occurrence of fog and mist exhibit negative trends in the months with the highest share, January and February.*

Also, there are unnecessary information throughout the main text that should be remove to make the manuscript clearer. For example: - L.75-80 are not necessary - Same on L.98 (information on population) - L.439-441 (or rephrase and include in the introduction)

Reply: The selected parts were removed from the manuscript.

**Comment 2.2.** The choice of 9 best matching units (BMUs) needs to be discussed more. This choice seems to follow Matić et al. (2022, who did not discuss this choice either). Also, Matić et al. (2022) were working on September only, while the current study applies the Growing Neural Gas (GNG) to six months. So the question of a fixed number of BMUs for different months rises question and should be discussed.

Reply: The discussion of BMUs will be improved according to the suggestions.

The reviewer is right, the GNG analysis is based on the work of Matić et al. (2022), in which it was done only for September, whereas in this paper it was done for all 12 months - each time for each month separately, but only 6 selected months were analyzed with more details, which we will discuss later.

The number of 9 BMUs was chosen as an optimal number and the best dimensionality reduction we can get in terms of interpretation. Considering that we are working with a complex and highly variable data set, our goal was to capture as many patterns as possible. However, we did not want to be too coarse and only select a few BMUs. At the same time, we also did not want to take too many BMUs, as the aim of the GNG analysis is to simplify and reduce the dimensionality of the data. We need to be able to describe and interpret BMUs, but at the same time we must not overload the work as analyzing and writing about 50 BMUs is impractical. The more BMUs we select, the fewer errors we get, which is logical, but then the question of the purpose of dimensionality reduction arises. In a previous work published by one of the authors of this paper, the selection of the number of BMUs was based on an assessment that focused on dimensionality reduction (for example Šantić, D., Stojan, I., Matić, F. et al. Picoplankton diversity in an oligotrophic and high salinity environment in the central Adriatic Sea. Sci Rep 13, 7617 (2023), https://doi.org/10.1038/s41598-023-34704-9; Skejić S, Milić Roje B, Matić F, Arapov J, Francé J, Bužančić M, Bakrač A, Straka M, Ninčević Gladan Ž. Phytoplankton Assemblage over a 14-Year Period in the Adriatic Sea: Patterns and Trends. Biology (Basel). 2024 Jul 2;13(7):493. doi: 10.3390/biology13070493.).

The reason why the number of 9 BMUs was chosen for each month is that it might be helpful to stay with a consistent number of BMUs for each month's data, as we analyze the data on a monthly basis.

For example, different quality measurements are available to assess the clustering obtained from the self-organizing maps (SOM), such as the quantization error, the topographic error or the percentage of explained variance (Bauer et al., 1999; Kaski and Lagus, 1996; Kohonen, 2001; Kohonen et al., 2009). So it could be possible to use similar measurement with the GNG to select the best number of BMUs for each month. Regarding the quality measures Matić et al. (2018) writes that none of these measures correspond to the usual measures used in the quality assessment of data mapping in the geosciences, where bias and root-mean-square error are normally used to assess the goodness of fit of a model to the data (Frano Matić, Hrvoje Kalinić, Ivica Vilibić, Interpreting Self-Organizing Map errors in the classification of ocean patterns,Computers & Geosciences,Volume 119,2018,Pages 9-17,ISSN 0098-3004,https://doi.org/10.1016/j.cageo.2018.06.006).

And another reason for the simple BMUs interpretation is that the GNG analysis was carried out on a large server because it was performed with hourly data from the ERA5 reanalysis, which, when including 12 separate months spanning 40 years and averaging 30 days by 24 hours by 2 (u and v components) by 73 x 193 (horizontal resolution), results in a huge data set. Server time had to be reserved for this dataset, which we were unable to obtain at the time of this review. Therefore,

unfortunately, we were not able to perform additional computational analyses such as calculating the quantization error to find a new number of correct BMUs to respond to this review.

Also, please add a short supplementary text to explain more what "the method outlined in Matić et al. (2022)" (L.182) is and how it has been adapted to the current work.

The part of the manuscript describing the method has been expanded to better explain the adaptation of the method to the current work.

Old version:

*In this study, adapted method outlined in Matić et al. (2022) was applied to a high-dimensional, 40-year dataset with wind components u and v for the entire year. Nine BMUs per month were calculated for each month of the year using the GNG algorithm from the Python library NeuPy. Data processing and visualization were performed using Python and MATLAB, with MATLAB employing the mapping package M_map (Pawlowicz, 2020) which is available online at www.eoas.ubc.ca/~rich/map.html.*

Was replaced with this:

*In this study, the methodology described in Matić et al. (2022) was applied to a high-dimensional, hourly, 40-year dataset comprising wind components u and v, analyzed separately for each month of the year. Prior to implementing the GNG algorithm, the wind data was arranged in such a way that two columns represented the spatial variations of u and v, while rows corresponded to temporal instances. Wind data over land was excluded and assigned as NaN values to reduce calculation time. This exclusion did not affect the results, as the study's primary focus was on the synoptic-scale influences that generate specific wind patterns conducive to the formation of fog and mist. Over the sea, the winds are very homogeneous and are often directly related to large-scale synoptic systems such as cyclones or anticyclones, whereas over land, the influence of topography and vegetation introduces noise, masking these processes. The proximity of the airfield to the sea and the flat, low-lying terrain in the direction of prevailing winds (Figure 1b) justified this simplification of calculations. The GNG algorithm, implemented using the NeuPy Python library and parameterized following Matić et al. (2022), was used to calculate 9 BMUs per month for each month of the year. These BMUs were then linked to corresponding mean sea level pressure (MSLP) fields. The temporal sequence of BMUs was utilized to compute an average pressure field for each BMU, offering a detailed representation of atmospheric patterns associated with fog formation.*

*Data processing and visualization were conducted using Python and MATLAB, with MATLAB employing the M_map mapping package (Pawlowicz, 2020), available at www.eoas.ubc.ca/~rich/map.html.*

The authors say that "[...] the GNG algorithm was applied only to 10-m wind data from ERA5, and the derived pressure fields were subsequently extrapolated" (L.316-318). What is a derived pressure field? And why is it extrapolated? Also, removing the MSLP from the analysis is an adaptation of the original work by Matić et al. (2022), which needs to be described either in the main text or as a supplementary text.

Reply: The reviewer is correct; this is unclear without further clarification.

A large part of this issue was addressed in the previous response, in which the reviewer asked us to expand on the description of the methodology. In this way, by expanding the description of the methodology, this question was also answered. Accordingly, the text has been revised, part of it has been moved to the methods, and part of it has been left as is.

The derived pressure field represents the mean pressure field calculated for each BMU. To reduce the calculation time, only wind data was entered into the GNG algorithm. Once the BMUs were identified, all pressure fields corresponding to the times at which each BMU occurred were retrieved. These pressure fields were then averaged to determine the mean pressure field for each BMU. The physical reason was that wind and pressure are closely related and we can easily reconstruct pressure fields based on the wind.

A small note, the method has not been changed if we look at the description of the work by Matić et al. (2022):

*By classifying wind speeds over the Mediterranean Sea and western Atlantic Ocean using GNG, we could identify the average and extreme wind patterns responsible for variations in the oceanographic features of the Adriatic Sea (Figure 5). With GNG, the wind field was classified into nine clusters and the average MSLP was calculated for each cluster. Based on the wind speed and MSLP patterns, the meteorological conditions are described as follows.*

Old version of the text:

*The reason for this approach is that it makes the results easier to interpret, captures seasonality, and reduces the computational load, which is important given the large hourly dataset involved. To save computational resources, the GNG algorithm was applied only to 10-m wind data from ERA5, and the derived pressure fields were subsequently extrapolated.*

New version of the text:

*This approach was chosen to improve the interpretability of the results, to take seasonality into account and to reduce the computational effort, especially given the large size of the hourly dataset. To save resources, the GNG algorithm was applied exclusively to 10 m wind data from ERA5, with the derived pressure fields being extrapolated afterwards. The derived pressure field corresponds to the mean pressure field associated with each BMU. Once the BMUs were identified, all pressure fields corresponding to the time instances of each BMU were retrieved and averaged to calculate the mean pressure field for each BMU.*

The selections described in L.337-341 are not scientifically based and must be discussed. Particularly, the study states that GNG is applied to all the months of the year, to finally focus to only half of them. One solution could be to use Fig.2b (with the changes suggested below) to get rid of the arbitrary choice.

Reply: The reviewer is correct. GNG was applied to all months at the beginning of the writing, but during the analysis it was decided to select only half of the months, based on an arbitrary selection. We have retained this part of the text, which states that the GNG analysis was carried out for all months to see the amount of work done, which in reality was not necessary as half a year could have been omitted. In accordance with the author's instructions, we have removed the arbitrary selection and revised the text so that it has a stronger scientific basis based on Figure 2b. In this way, we have captured the months that show 85% of the variability.

A new criterion was introduced to eliminate the arbitrariness of BMUs. The first BMUs whose sum of their shares of variability exceeded the threshold of 60% were selected. In this way, 18 BMUs were selected for further analysis, which is significantly fewer than if we were to consider all 54 BMUs (6 months multiplied by 9 BMUs), which facilitating further scientific analysis. We can therefore say that we always captured more than 60 of the events for each analyzed month. Furthermore, additional analysis at the reviewer's instigation found that BMU-2-7 for February, which has high frequency, was omitted and therefore it was inserted into the manuscript. Changes have also been made to the main text because of that, namely in the section describing the results and in the discussion and conclusions.

Old version of text:

*To focus on the prevailing synoptic patterns that contribute to the formation of fog and mist, months with more than 10% of days with fog and mist were arbitrarily selected (Table 1). These months are in the colder part of the year and together account for more than 85% of the foggy days of the year. A criterion relating to BMUs was also arbitrarily defined for each month: Only BMUs with more than 15% representation in a given month were selected (Table 1) and plotted (Figures 9-14). In this way, 2 to 4 BMUs were selected and more closely analyzed for each month.*

Revised text:

*To focus on the prevailing synoptic patterns that contribute to the formation of fog and mist, the months from October to March (Figure 2b) were examined in more detail (Table 1). In this way, 85 % of the variability could be captured. Then, for each month, the first BMUs whose sum of their contributions to the variability exceeded the threshold of 60 % were selected. In this way, a total of 18 BMUs were included in the further analysis, which is significantly fewer than 54 BMUs (6 months multiplied by 9 BMUs). By selecting 2 to 4 BMUs for each month based on the criteria described above, the study captured more than 60 percent of the events for each analyzed month.*

**Comment 2.3.** The comment "[...] local variations such as topography or land features can have a large influence on the results." (L.330) contradicts the statement "[…] considering the specific terrain and coastline features..." (L.93). Displaying the arrows over land is important to capture the regional patterns of atmospheric circulation near the surface. Please include the wind over land.

Reply: The reviewer is correct. We were thinking more about general synoptic features that generate the wind that can lead to the fog and mist events on Pula, and not so much about the variation of the wind on land. Therefore, these sentences are omitted from the text to make the discussion clearer and more complete.

Another reason why we cannot include the wind over land is mentioned in the answers above and is also related to the fact that we cannot get further processing time for a new analysis within the review period, which would take a lot of time. We also repeat the answer from earlier, in which we gave a high priority to synoptics because we were mainly concerned with climatology in this paper:

*Wind data over land was excluded and assigned as NaN values to reduce calculation time. This exclusion did not affect the results, as the study's primary focus was on the synoptic-scale influences that generate specific wind patterns conducive to the formation of fog and mist. Over the sea, the winds are very homogeneous and are often directly related to large-scale synoptic systems such as cyclones or anticyclones, whereas over land, the influence of topography and vegetation introduces noise, masking these processes. The proximity of the airfield to the sea and the flat, low-lying terrain in the direction of prevailing winds (Figure 1b) justified this simplification of calculations.*

For some BMUs, the orientation of the arrows looks different from the wind origin reported in Table 1. For example, the wind of BMU-1-8 is WNW/NW, but in the associated map, the main flux over the Istrian Peninsula is more SW. Same for BMU-3-1. Please check. Also, for some panels, there is a discrepancy between the contour and the orienation of the arrows (e.g., BMU-3-1, BMU-3-3, BMU-12-1). Please check.

In accordance with the advice of reviewer number 1 and the note of reviewer number 2 to make an additional comparison of the wind direction in the table and in the figures, a zoomed panel of maps was created showing the area of the northern and central Adriatic (Figures 9, 10, 11, 12, 13). Looking at the new maps, it was determined that the wind was well written in table number 1 for BMU-1-8 and BMU-3-1.

In order to check if there is a discrepancy between the contour and the orientation of the arrows for BMU-3-1, BMU-3-3, BMU-12-1, new figures were created containing only BMU-3-1 (Figure 1), BMU-3-3 (Figure 2) and BMU-12-1 (Figure 3). If we look at these diagrams, we can see that the wind direction around the cyclones is counterclockwise (Figure 1 – Cyprus, Figure 2 – Northern Italy, Figure 3 – Israel). So we can say that the result is physically based. Small deviations between the contour and the orientation of the arrows may occur because for each BMU we have averaged all pressure fields that occur when this BMU occurs in time. In addition, the wind is not influenced by a pure pressure gradient and is not completely parallel to the isobaths, but there is also the influence of Coriolis, the centrifugal force...

[Figure]

**Figure 1. BMU-3-1 describes the synoptic patterns favoring the formation of fog and mist at Pula Airport in March for the wider Mediterranean region). The red dot marks the location of Pula Airport. The red contour lines show the mean sea level pressure (MSLP). The contours represent the amplitude of the wind speed, above which the wind vectors are represented by arrows (every third vector has been drawn).**

[Figure]

**Figure 2 BMU-3-3 describes the synoptic patterns favoring the formation of fog and mist at Pula Airport in March for the wider Mediterranean region). The red dot marks the location of Pula Airport. The red contour lines show the mean sea level pressure (MSLP). The contours represent the amplitude of the wind speed, above which the wind vectors are represented by arrows (every third vector has been drawn).**

[Figure]

**Figure 3 BMU-12-1 describes the synoptic patterns favoring the formation of fog and mist at Pula Airport in December for the wider Mediterranean region). The red dot marks the location of Pula Airport. The red contour lines show the mean sea level pressure (MSLP). The contours represent the amplitude of the wind speed, above which the wind vectors are represented by arrows (every third vector has been drawn).**

**Comment 3.1.** Figure 1: Please move labels "a)" and "b)" at the top of each panel.

Reply: The labels were moved.

What is the meaning of "20" and "24" in the legend? If not necessary, please delete.

Reply: This is a typo. It was supposed to refer to 2024 in the copyright exclamation (c Google Maps 2024), but it was accidentally mistyped in the wrong line. This was corrected.

a) The bathymetry is not necessary, please simplify the information. Also, the inset should not go beyond the edges of the upper panel.

Reply: The inset was corrected and the bathymetry removed.

Figure 2b: It is better to center the barplot on boreal winter, so we have a clearer view. And this will help for the choice of the months the GNG is applied to (see comment above). Also, the legends are overlapping, please fix.

Reply: The barplot was centered on the boreal winter and the legends were corrected.

Figures 5 and 6: What it the meaning of "VRB"?

Reply: VRB refers to variable wind in aeronautical meteorology: variability of 180° or more in a 2-minute interval according to definition by ICAO (International Civil Aviation Organization). The captions were revised and they now include this information.

Figure 9: Reshape the panels to have January (February) in top (bottom) row. Also, it is better to have the label above each panel, rather than on the right. Finally, please add in the caption the meaning of the arrows, contours and shading.

Reply: Most of the reviewer's comment was accepted and there is a new caption:

*Figure 9. Prevailing Best Matching Units (BMUs) describing the synoptic patterns favoring the formation of fog and mist at Pula Airport in January (a, c) and February (e, g) for the wider Mediterranean region. The red dot marks the location of Pula Airport. The red contour lines show the mean sea level pressure (MSLP). The contours represent the amplitude of the wind speed, above which the wind vectors are represented by arrows (every third vector has been drawn). The same principle applies to the zoomed area of the northern and central Adriatic for January (b, d) and February (f, h), but here each vector is plotted.*

However, due to the request of another reviewer to include a more detailed description of the northern Adriatic region, which was done, not all guidelines could be accepted so an optimal compromise was sought.

Figure 10–13: Please move the labels above the panels. Also, please simplify the figure captions by referencing to the Fig. 9.

Reply: The suggestions were accepted and the figure revised.

Figure 14: Add labels (a)-(q) and use in the main text.

Reply: The corrections were made accordingly.

**Comment 3.2**. Table 1: "The slope coefficients describe the linear trends of the most common BMUs, i.e. the yearly change in frequency". Is it related to Fig.14? If yes, please add a reference to it.

Reply: Yes, it's related and the reference was added to the manuscript.

**Comment 3.3.** The title is rather long. Authors could consider to shorten it (Deng, 2015).

Reply: We propose to shorten the title to: *Statistical and neural network assessment of the climatology of fog and mist at Pula airport in Croatia*

L.90: Citation is incorrect/incomplete.

Reply: Citation has been corrected in the reference list.

L.146: What is the meaning of "METAR" and "SYNOP"?

Reply: METAR (METeorological Aerodrome Report) is a coded report describing weather conditions at the airport in a manner standardized for aviation. SYNOP (Surface Synoptic Observations) is a coded report describing weather conditions at a meteorological station. An airport meteorological station sends both SYNOP and METAR reports. This explanation was added to the manuscript.

L.161: What is the meaning of "level 4"?

Reply: The L4 data is a gridded analysis product, i.e. the data has been processed, analyzed and interpolated to close gaps and ensure continuous and consistent coverage. To simplify the manuscript, the phrase was removed and the term *gap-free* was introduced.

Old text:

*This dataset, which has been optimally interpolated (level 4) with a grid resolution of 0.05° (Merchant et al., 2019), provides a comprehensive view of sea surface temperatures.*

New text:
*This gap-free dataset, which has been optimally interpolated with a grid resolution of 0.05° (Merchant et al., 2019), provides a comprehensive view of sea surface temperatures.*

L.174: Why this region has been chosen? Is it based on Matić et al. (2022), who used the same region in their study?

Reply: In answering the reviewer's question, an error was found in the original text:

*The covered area extends from 20°W to 40°E and 25°N to 55°N (Figure 1a, smaller map), with data spanning from 1979 to 2019.*

However, a look at the maps shows that the covered area extends from 6°W to 42°E and 30°N to 48°N.

Compared to Matić et al. (2022) "...20 W–40 E and 25 N–55 N coverage area...", this area is smaller. The reason for selecting this area was to cover an area that has the greatest synoptic influence on the Adriatic Sea in terms of wind regime, such as the Genova cyclone or the anticyclone over southeastern Europe, but at the same time not to extend the area too much, thus slowing down the computation for the GNG analysis.

New revised text:

*The study area spans from 6°W to 42°E and 30°N to 48°N (Figure 1a, inset map), encompassing the region where key synoptic processes influencing the Adriatic Sea predominantly occur. The analysis covers a 40-year period from 1979 to 2019, providing a comprehensive temporal dataset for assessing atmospheric conditions. At the same time, this area optimizes the calculation time.*

L.203-205: Please use the value of the slope to describe the trend.

Reply: The trend was described as the reviewer suggested.

Old text:
*A careful evaluation of the linear trend reveals that the average number 203 of fog days has decreased by more than 10 days (from 18.4 in 2001 to 8.3 in 2020), and the average number of mist days has decreased by more than 22 days (from 56.8 in 2001 to 34.4 in 2020).*

New text:
*A detailed evaluation of the linear trend indicates that the average number of fog days decreased by more than 10 days, from 18.4 in 2001 to 8.3 in 2020, with a slope coefficient of -0.53. Similarly, the average number of mist days declined by over 22 days, from 56.8 in 2001 to 34.4 in 2020, with a slope coefficient of -1.18. The larger absolute value of the slope coefficient for mist suggests that the average number of mist days is declining at a faster rate.*

L.296: "[…] confirmed by calculations". What does it mean?

Reply: This refers to statistical analyses performed to obtain the SST variance. This was corrected and rephrased to make it clearer.

Old text:

*This is confirmed by calculations, which show that variance for SST of the western area (for observations with fog) is 12.62 °C, and for the eastern area, it is 11.37 °C. The same is true for observations with mist (16.24 and 14.66 °C, respectively). Further insight into the influence of SST on fog formation was gained by separately analyzing the differences between the SST of the western and eastern areas in cases with and without fog or mist (Figure 8).*

New text:

*This was confirmed by statistical analyses, which show that the standard deviation for SST of the western area (for observations with fog) is 3.55 °C, and for the eastern area, it is 3.37 °C. The same is true for observations with mist (4.03 °C for the western area and 3.83 °C for the eastern area). To better understand the role of sea surface temperature (SST) in fog formation, we analyzed the differences in SST between the western and eastern areas under conditions with and without fog or mist (Figure 8).*

L.312: It is not clear what the author mean by "to extract characteristic temporal and spatial patterns". Please clarify.

Reply: Rephrased to make it clearer.

New text:

*In this analysis, the GNG method was applied to identify characteristic temporal and spatial patterns in wind and MSLP fields that are associated with the formation of fog and mist in the Pula region.*

L.326-328: Not sure about the meaning of the sentence. Could the authors explain further?

Reply: Rephrased to make it clearer:

Old text:

*Given that wind was the primary variable and MSLP the derived variable, wind exerted the predominant influence, thus excluding the occurrence of very strong cyclonic (MSLP < 1000 hPa) or anticyclonic (MSLP > 1030 hPa) formations in the results. However, from a conceptual point of view, and with regard to the dynamic interaction between high- and low-pressure systems, the results are satisfactory and align well with the climatology of the region.*

New text:

*Since the wind was the primary variable and the MSLP was derived by averaging numerous synoptic situations, the wind had the dominant influence. This approach excluded the occurrence of extreme cyclonic systems (MSLP < 1000 hPa) or anticyclonic (MSLP > 1030 hPa) from the results. For example, deep cyclones, which are extremes, are smoothed, and while BMUs can indicate their locations, the exact pressure values are not retained. However, from a conceptual point of view, the interaction between high and low pressure systems and the location of these systems and their associated winds match well with the climatology of the region.*

L.380-381: What mechanism(s) are in action?

Reply: The text was revised accordingly.

Old text:

*The intensified synoptic pressure gradients over the central Mediterranean contribute to increased NE wind patterns. This increased wind activity can result in moist air being transported from the sea to the coastal regions, providing an additional source of moisture for fog formation. The convergence of air masses along these enhanced pressure gradients can also promote upward movements, leading to local cooling and increased humidity.*

New text:

*The intensified synoptic pressure gradients over the central Mediterranean contribute to increased NE wind patterns. This increased wind activity can result in moist air being transported from the sea to the coastal regions, providing an additional source of moisture for fog formation. The convergence of air masses along these enhanced pressure gradients likely induces upward motion of air, which can result in adiabatic cooling and an increase in relative humidity, creating conditions favorable to fog and mist formation.*

L.395-396: "cyclonic conditions" rather than "cyclone" maybe?

Reply: Revised accordingly.

L.469: "VRB" must be defined earlier.

The explanation was added to the manuscript, please refer to previous reply to comment in section 3.1.

L.513: There was not previous mention of global warming/climate change and their effect on the frequency of fog/mist days. This should be mention in the introduction.

Reply: The reference to climate change was added to the introduction:

*In addition to local factors influencing fog and mist formation, the impact of global warming and climate change cannot be overlooked. These global phenomena have been linked to a reduction in the number of days with fog and mist (Kawai et al., 2016; Klemm and Lin, 2016).*

L.557-674: Please provide the article titles in sentence case.

L.575: Journal abbreviation.

L.586: Please supply the full author list with last name followed by initials. (https://www.nonlinear-processes-in-geophysics.net/submission.html#references).

L.589: Same as L.586 comment.

L.591: Same as L.586 comment.

L.616: Incorrect title of the paper.

L.618: Same as L.586 comment.

L.620: Same as L.586 comment.

L.621: Journal abbreviation.

L.670: Same as L.586 comment.

Reply: The references were corrected.

---

## Referee Report (RR1)

This study first statistically analyzed the atmospheric conditions related to the fog and mist. Then, they used machine learning to analyze the prevailing patterns associated with the fog and mist. It will be useful for predicting fog and mist and is worthy of publication.

All my previous comments were addressed by the authors well. However, the conclusion part and some other parts need to be improved before publication.

**About the conclusion**

The conclusion is still not well written. The conclusion should summarize the authors' main findings in the paper, not providing too much new information. For example,

Line 493-498

This paragraph explains the decreasing fog and mist trend and provides some references. It should be moved to the main text. In the conclusion, only use a few sentences to repeat what the authors found.

Line 500 – 508

This is how global warming impacts the SST and then fog and mist. They should be moved to the main text, and a few sentences should be used here to summarize.

The main findings from Figure 4-13 were not summarized in the conclusion. Please use a few sentences to summarize the main findings from Figure 4-13.

Line 510-516

This is related to Figure 14 but too much content. They should be moved to the main text. The authors can use a few sentences here to summarize Figure 14.

**Other issues:**

Line 168

From here, the authors started to describe the GNG method. It is better to start a new paragraph to improve readability.

Line 184

" whereas over land, the influence of topography and vegetation introduces noise, masking these processes."

I don't understand 'masking these processes.' what processes? And why 'masking'
Maybe it is better to use another word.

Line 207
"Airport, 2021-2020."
Should it be 2001-2020?

Line 302
"Fog rarely occurs in calm conditions, suggesting an optimal wind speed range for its formation and warranting further exploration, a "

Which figure justifies this sentence? Figure 6b shows dots when the wind speed is 0. It looks like fog can also occur in calm conditions.

Line326 Figure 7
It is optional. The authors can consider adding a diagonal line to indicate where SST = SAT.
So the reader can better visualize the parts where SST > SAT and SST < SAT.

Line 329 Figure 8 caption
'without fog or mist'

Please confirm if it is 'Without fog **or** mist" or ''without fog **and** mist'
'Without fog **or** mist" means at least one (or both) of these conditions is absent.
"Without fog and mist" means the complete absence of both fog and mist.
Since panels b and c are fog and mist. It looks like panel a means 'without fog **and** mist'.

Line 420
"figure 13g, 13h"
I think it should be "figure 13e, 13f".   There are cyclones in Figures 13e and 13f.

Line 506
"Warmer SSTs reduce the temperature gradient required for fog formation and increase evaporation rates, promoting fog advection when winds are favorable."

I think warmer SST increases the temperature gradient between SST and air temperature above if SST > SAT. Please explain.

---

## Author Response (AR2)

Zagreb, 8 February 2025

Editorial Team

Nonlinear Processes in Geophysics

Dear Editor,

Please find enclosed a revised version of the manuscript titled "*Statistical and neural network assessment of the climatology of fog and mist at Pula airport in Croatia*", revised according to additional suggestions from Reviewer #1.

We have provided two versions of the manuscript: one with tracked changes in Word, and the other with normal text, and additionally all the changes are also referenced in this document. Below, we present the reviewer's comments in regular text, while our responses are highlighted in yellow.

On behalf of myself and all co-authors,

Yours sincerely,

Marko Zoldoš

**Comment 1.** The conclusion is still not well written. The conclusion should summarize the authors' main findings in the paper, not providing too much new information. For example,

Reply: The conclusion was rewritten according to the reviewer's suggestions. Details are below:

**Comment 2.** Line 493-498: This paragraph explains the decreasing fog and mist trend and provides some references. It should be moved to the main text. In the conclusion, only use a few sentences to repeat what the authors found.

Reply: The paragraph was moved to Chapter 3.1 and replaced with a shorter summary.

**Comment 3.** Line 500 – 508: This is how global warming impacts the SST and then fog and mist. They should be moved to the main text, and a few sentences should be used here to summarize.

Reply: The paragraph was moved to Chapter 3.1 and replaced with a shorter summary.

**Comment 4.** The main findings from Figure 4-13 were not summarized in the conclusion. Please use a few sentences to summarize the main findings from Figure 4-13.

Reply: Several new sentences were added to the conclusion to summarize these findings.

**Comment 5.** Line 510-516: This is related to Figure 14 but too much content. They should be moved to the main text. The authors can use a few sentences here to summarize Figure 14.

Reply: The paragraph has been deleted because the majority of the content is already described in the main text (Figure 14). It was replaced with a shorter summary.

**Comment 6.** Line 168: From here, the authors started to describe the GNG method. It is better to start a new paragraph to improve readability.

Reply: A new paragraph was started.

**Comment 7.** Line 184: "whereas over land, the influence of topography and vegetation introduces noise, masking these processes." I don't understand 'masking these processes.' what processes? And why 'masking'? Maybe it is better to use another word.

Reply: The reviewer is correct; the phrasing was unclear. The text has been rephrased:

*Over the sea, the winds are very homogeneous and are often directly related to large-scale synoptic systems such as cyclones or anticyclones. Over land, the influence of topography and vegetation introduces noise, making it more difficult to distinguish synoptic influences from local influences.*

**Comment 8.** Line 207: "Airport, 2021-2020." Should it be 2001-2020?

Reply: Yes, this was a typo which was corrected.

**Comment 9.** Line 302: "Fog rarely occurs in calm conditions, suggesting an optimal wind speed range for its formation and warranting further exploration, a "

Which figure justifies this sentence? Figure 6b shows dots when the wind speed is 0. It looks like fog can also occur in calm conditions.

Reply: While fog can form in calm conditions, Figure 4b shows that it is rare compared to other categories (when wind speeds are 1 ms$^{-1}$ or lower). The color scale in Figure 4b has been improved to make this more visible, and the reference to the figure has been added to the text:

*Fog rarely occurs in calm conditions (Figure 4b), suggesting an optimal wind speed range for its formation and warranting further exploration, as the role of wind speed on turbulence and surface heat fluxes, as highlighted by Gultepe et al. (2007), significantly influences fog.*

**Comment 10.** Line 326, Figure 7: It is optional. The authors can consider adding a diagonal line to indicate where SST = SAT. So the reader can better visualize the parts where SST > SAT and SST < SAT.

Reply: The diagonal line was added to the figure, with an explanation in the captions. Furthermore, the figure has been slightly altered to better conform to rules regarding readers with color vision deficiencies.

**Comment 11.** Line 329, Figure 8 caption: 'without fog or mist'

Please confirm if it is 'Without fog or mist" or ''without fog and mist'. 'Without fog or mist" means at least one (or both) of these conditions is absent. "Without fog and mist" means the complete absence of both fog and mist. Since panels b and c are fog and mist. It looks like panel a means 'without fog and mist'.

Reply: Yes, "without fog and mist" is the correct phrase, and the caption was corrected.

**Comment 12.** Line 420, "figure 13g, 13h": I think it should be "figure 13e, 13f". There are cyclones in Figures 13e and 13f.

Reply: Yes, "figure13e, 13f" are the correct figure titles. The text was corrected.

**Comment 13.** Line 506: "Warmer SSTs reduce the temperature gradient required for fog formation and increase evaporation rates, promoting fog advection when winds are favorable."

I think warmer SST increases the temperature gradient between SST and air temperature above if SST > SAT. Please explain.

Reply: We considered only those specific cases where SST<SAT, but the sentence was indeed unclear. It was rephrased (and also moved from the conclusion to the main text):
*Warmer SSTs influence fog formation in two ways. Generally, they increase evaporation rates, enhancing fog advection when winds are favorable. Additionally, in cases where fog forms with SST < SAT, they reduce the temperature gradient required for fog development.*

---

## Author Response (AR3)

Zagreb, 15 February 2025

Editorial Team

Nonlinear Processes in Geophysics

Dear Editor,

On behalf of my co-authors and myself, I would like to sincerely thank you for accepting our paper for publication in *Nonlinear Processes in Geophysics*. We also extend our gratitude to the reviewers for their constructive suggestions and comments, which have helped improve the quality of our manuscript.

Please find enclosed the final revised version of our manuscript, titled "*Statistical and neural network assessment of the climatology of fog and mist at Pula airport in Croatia*". The only modification compared to the previously uploaded version is on lines 96–97, where we have updated the information about Pula Airport with the latest data from 2024. Additionally, we have decided to omit the supplementary materials previously uploaded, as the manuscript revisions have rendered them outdated and no longer of additional value.

On behalf of all co-authors, I sincerely appreciate your time and consideration.

Yours sincerely,

Marko Zoldoš